# AlphaFold-guided phylogenetic analyses suggest surprising heterogeneity in metazoan replication origin licensing mechanisms

Olivia Hunker & Franziska Bleichert 📧

## Abstract

**DNA replication initiation is a tightly regulated process that requires the coordinated assembly of replication machineries throughout the genome. During the first step of initiation, origin licensing, the MCM replicative helicase motor is loaded onto replication origins by the origin recognition complex (ORC) as a head-to-head double hexamer complex. Distinct mechanisms have been proposed to facilitate human MCM double hexamer loading, but the physiological relevance of each of them remains unclear. Here, we investigate the evolutionary conservation of these pathways using an AlphaFold-guided structural phylogenetics approach. Our analyses reveal that ORC6, a subunit of ORC previously thought to be essential for origin licensing in vivo, has been lost in multiple metazoan lineages. Despite this loss, many of these species retain an element in ORC3, the ORC3 tether, that can interact with MCM and facilitate an ORC6-independent MCM loading mechanism. AlphaFold2 Multimer predictions suggest that ORC3 tether interactions with MCM are broadly conserved across Metazoa. Our findings support the physiological relevance of ORC6-independent MCM loading, provide experimentally testable hypotheses on origin licensing mechanisms in diverse metazoan species, and highlight how AlphaFold can be leveraged to investigate protein evolution and function over large timescales.**

**Keywords** DNA Replication Initiation; Origin Licensing; Structural Phylogenetics; AlphaFold
**Subject Categories** DNA Replication, Recombination & Repair; Evolution & Ecology; Structural Biology

## Introduction

DNA replication is a fundamental process of life that is required for the accurate transmission of genetic information. Timely assembly of DNA replication machineries at origins is an essential prerequisite for this program to commence. In eukaryotes, replication initiation begins with the loading of the core of the replicative helicase, the heterohexameric MCM2-7 complex, onto replication origin sites during G1 in a step known as origin licensing, during which the origin recognition complex (ORC) sequentially recruits and loads two copies of MCM2-7 onto DNA to form a head-to-head double hexamer (Parker et al, 2017; Bleichert, 2019; Costa and Diffley, 2022). In S phase, DNA melting and unwinding at origins is triggered when MCM is activated by firing factors. During activation, the two MCM rings in the double hexamer disengage and begin translocating in opposite directions, forming bidirectional replication forks (Parker et al, 2017; Costa and Diffley, 2022).

Elucidating the molecular mechanisms governing origin licensing is critical for understanding genome maintenance and stability. Accordingly, models of eukaryotic MCM double hexamer loading have been proposed for both budding yeast and humans, based on in vitro biochemical, single-molecule, and structural studies (Evrin et al, 2009; Remus et al, 2009; Ticau et al, 2015; Yuan et al, 2017; Miller et al, 2019; Gupta et al, 2021; Yang et al, 2024; Weissmann et al, 2024; Wells et al, 2025). These findings have revealed a unified eukaryotic mechanism for loading of the first MCM hexamer onto DNA (Fig. EV1A). To accomplish this task, ORC binds to an origin site (Bell and Stillman, 1992) and then recruits the co-loader CDC6, which helps stabilize ORC on DNA (Seki and Diffley, 2000; Speck et al, 2005; Schmidt and Bleichert, 2020). An MCM hexamer, along with the co-factor CDT1, is recruited to the ORC-DNA-CDC6 complex through interactions between ORC and the C-terminal face of the MCM hexamer, resulting in an ORC-CDC6-CDT1-MCM (OCCM) intermediate in which DNA has been inserted in the MCM central pore (Yuan et al, 2017; Weissmann et al, 2024; Wells et al, 2025).

The task of loading the second MCM hexamer poses an additional challenge: it must be recruited in close proximity to and in opposite orientation of the first hexamer to successfully assemble the MCM double hexamer. In yeast, after dissociation of Cdc6 and Cdt1, ORC flips to the opposite N-terminal face of the first MCM hexamer, forming a second intermediate called the MO complex (Miller et al, 2019; Gupta et al, 2021; Driscoll et al, 2025). The MO positions ORC to recruit and load the second hexamer in the opposite orientation to the first to establish the correct head-to-head geometry of the MCM double hexamer. The yeast Orc6 subunit plays a key role in this transition by stabilizing the MO complex and by tethering ORC to MCM during the flip (Miller et al, 2019; Gupta et al, 2021; Amasino et al, 2023; Driscoll et al, 2025).

Department of Molecular Biophysics and Biochemistry, Yale University, New Haven, CT, USA. 📧E-mail: franziska.bleichert@yale.edu

Recruitment and loading of the second MCM hexamer in the human system does not strictly require the ORC6 subunit or the MO complex (Yang et al, 2024; Weissmann et al, 2024; Wells et al, 2025). Recent data have suggested that the human OCCM transitions to an MCM double hexamer (MCM-DH) first by dissociation of CDC6, CDT1, and ORC, resulting in a loaded MCM single hexamer (MCM-SH) (Yang et al, 2024; Weissmann et al, 2024). Next, the MCM-SH can mature into the MCM-DH through multiple, discrete mechanisms (Yang et al, 2024; Weissmann et al, 2024) (Fig. EV1A). The first one involves an ORC6-dependent MO-I intermediate, which, unlike the yeast MO, likely promotes MCM-DH formation by orienting uncoupled MCM-SH loading events to establish two loaded, N-terminally facing MCM-SHs that can then self-dimerize (Yang et al, 2024; Weissmann et al, 2024). The second pathway involves an MO-II intermediate that can form without ORC6 and relies on a region of the ORC3 subunit called the ORC3 tether, which binds to MCM2 (Yang et al, 2024). The conformation of the MO-II is compatible with direct recruitment of a second MCM hexamer to establish the MCM-DH as in yeast. A third pathway is MO-independent, requires neither ORC6 nor the ORC3 tether, and occurs when two independently loaded MCM-SHs self-associate on DNA, with the head-to-head orientation of MCM in each MCM-DH occurring by chance (Yang et al, 2024).

While biochemical studies support that human cells can use multiple pathways to form the MCM-DH, it remains unclear to what extent each of these routes are utilized in a physiological environment. Loading of the MCM-DH through MO-dependent mechanisms is likely advantageous because they allow specifying the orientation of each MCM hexamer with respect to the other, while MO-independent formation of the MCM-DH depends on stochastic alignment of correct hexamer orientations. Consistent with this line of reasoning, the ORC6 subunit and the ORC3 tether independently promote the efficiency of MCM-DH formation, with the presence of ORC6 having the largest impact (Yang et al, 2024). In U2OS cells, knockdown of ORC6 does not decrease the amount of MCM bound to chromatin; however, it is unknown if this MCM is double hexameric (Lin et al, 2022). Additionally, two missense mutations in ORC6 linked to Meier-Gorlin syndrome (MGS), a primordial dwarfism disease linked to DNA replication defects, specifically reduce MCM-DH formation through inhibition of MO formation (Yang et al, 2024). Regardless, MGS patient-derived cells with ORC6 mutations can still proliferate efficiently in culture, although S-phase progression is slowed (Stiff et al, 2013), and most mutations in ORC6 are associated with a milder MGS phenotype (Nielsen-Dandoroff et al, 2023). Thus, while experimental evidence suggests that some MCM-DH copies may be loaded in vivo without ORC6, the prevalence of this event remains uncertain. Additionally, to what extent each of the pathways to MCM-DH formation are conserved in other multicellular eukaryotes is unknown.

To address these questions, we conducted a structural phylogenetic analysis of protein interactions involved in MO-I and MO-II complex formation, guided by AlphaFold structural predictions across hundreds of metazoan species. Our findings reveal multiple independent losses of ORC6 across different metazoan lineages, accompanied by degeneration of the ORC6-binding domain in ORC3. By contrast, the ORC3 tether is broadly present, even in species lacking ORC6, suggesting an evolutionarily maintained mechanism for ORC6-independent MCM loading. Collectively, our results argue that ORC6-independent origin licensing occurs naturally in physiological environments and indicate that origin licensing mechanisms in multicellular eukaryotes are more diverse and adaptable than previously recognized.

# Results

## Multiple metazoan taxonomic groups lack ORC6

To explore the biological importance of the different MCM loading mechanisms in multicellular eukaryotes, we first analyzed the evolutionary conservation of ORC6. The five core subunits of ORC (ORC1-5) belong to the AAA+ ATPase superfamily (Neuwald et al, 1999; Iyer et al, 2004) and form a stable, domain-swapped pentameric ring that binds DNA in an ATP-dependent manner (Bell and Stillman, 1992; Chesnokov et al, 2001; Vashee et al, 2003; Bleichert et al, 2015; Bleichert et al, 2018; Li et al, 2018). ORC6 has a distinct lineage from ORC1-5, sharing structural homology with TFIIB and containing two cyclin-box folds ($CB_C$ and $CB_N$) along with a C-terminal domain (CTD) (Chesnokov et al, 2003; Liu et al, 2011; Bleichert et al, 2013; Xu et al, 2020) (Fig. 1A). A conserved α-helix in the ORC6-CTD associates with its binding domain (ORC6-BD) in ORC3, which itself is comprised of three α-helices arranged in a triangular pattern (Bleichert et al, 2013; Bleichert et al, 2015; Li et al, 2018) (Fig. 1A,B). In some eukaryotes, namely yeast and *Drosophila*, ORC6 stably binds to the ORC6-BD in ORC3 (Bleichert et al, 2013; Bleichert et al, 2015; Li et al, 2018); however, in *Xenopus* and humans, ORC6 is more loosely associated with the ORC1-5 core, binds ORC3 with lower affinity, and does not efficiently co-purify with the other subunits (Dhar and Dutta, 2000; Dhar et al, 2001; Gillespie et al, 2001; Vashee et al, 2001; Vashee et al, 2003; Giordano-Coltart et al, 2005; Ranjan and Gossen, 2006; Siddiqui and Stillman, 2007; Bleichert et al, 2013). These findings raise the possibility that ORC6 and its role in origin licensing are less conserved than the ORC1-5 subunits and their respective functions. Our curiosity was further driven by the striking absence of an identified ORC6 ortholog in the nematode *Caenorhabditis elegans*, a well-characterized model organism (Yang et al, 2024). These observations led us to postulate that the ORC6 subunit may not be conserved in all metazoan species.

To investigate this hypothesis, we conducted BLAST searches for orthologs of all ORC subunits across the metazoan kingdom using the human sequences as search queries. We inferred the conservation of each ORC subunit based on the presence or absence of orthologs across taxonomic groups. Phyla were excluded from analysis only if orthologs for more than one ORC subunit were entirely absent across all species with available sequence data in that phylum. Given the overrepresentation of certain metazoan phyla in genomic databases such as chordates and arthropods, we further divided these groups into subphyla to better account for their genetic diversity. Across all analyzed phyla and subphyla, we identified orthologs of each core ORC1-5 subunit in at least one species, consistent with their strong evolutionary conservation (Table EV1). As many genome assemblies are incomplete (Hotaling et al, 2021), we avoided interpreting absence at the species level as definitive. However, ORC6 orthologs were strikingly absent from the NCBI database for all species in multiple taxonomic groups, including Tunicata (a subphylum of Chordata), Rotifera, Platyhelminthes (flatworms), Nematoda, and Chelicerata (a subphylum of

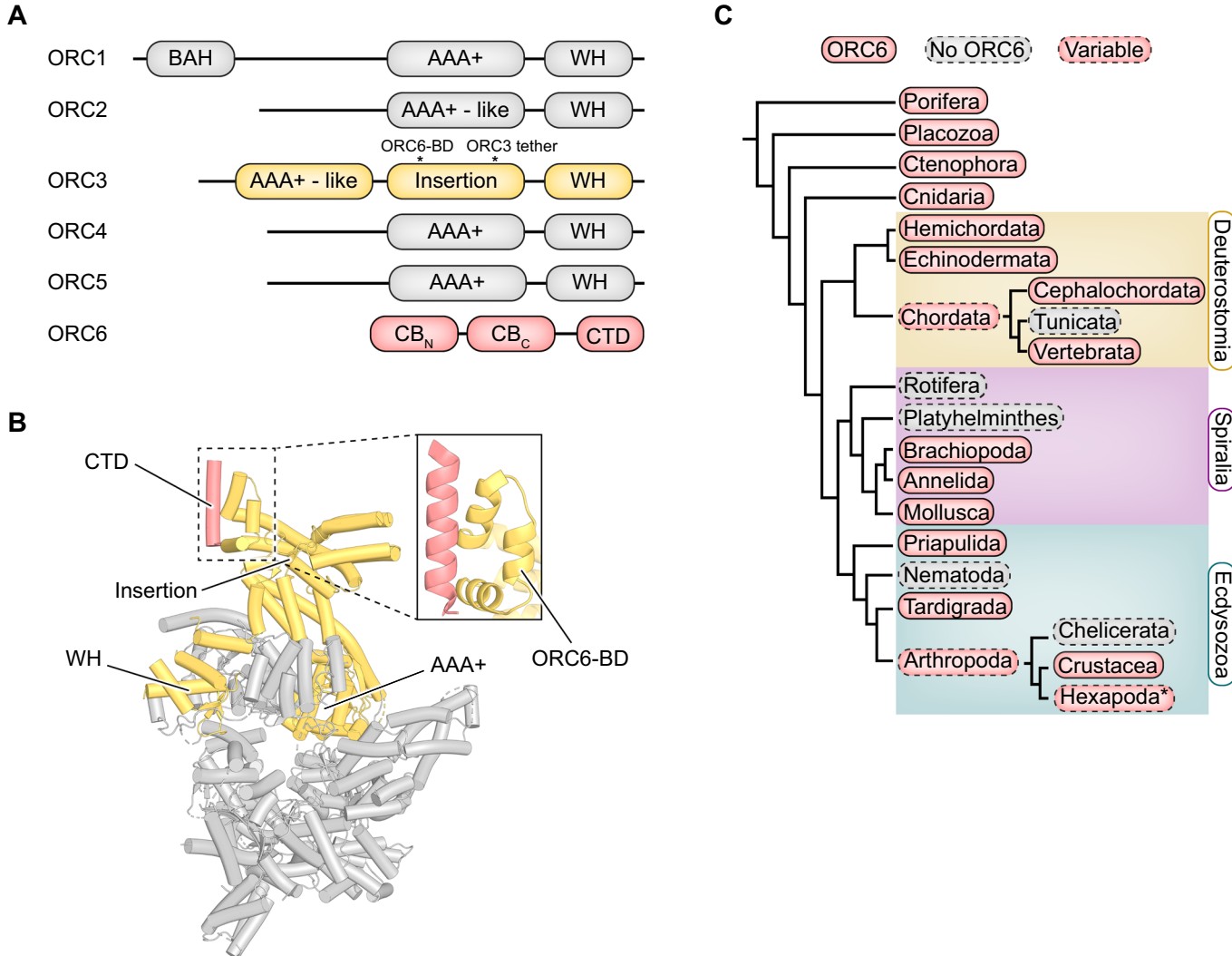

**Figure 1. Loss of ORC6 in a subset of Metazoa.**

(**A**) Domain architecture of ORC1-6 subunits. (**B**) Crystal structure of *Drosophila* ORC (PDB 4XGC (Bleichert et al, 2015)) illustrating the binding interface between the ORC6 (salmon) and ORC3 (yellow) subunits. (**C**) Phylogenetic tree of ORC6 conservation among metazoan phyla with identified ORC orthologs. Absence of ORC6 was inferred if ORC1-5 orthologs but no ORC6 ortholog were reliably identified through BLAST searches. Tree construction was based on widely accepted metazoan phylogeny (Laumer et al, 2019), and distances of branches depicted are arbitrary. BAH bromo-adjacent homology domain, WH winged helix domain, $CB_N$ N-terminal cyclin-box fold, $CB_C$ C-terminal cyclin-box fold, CTD C-terminal domain, ORC6-BD ORC6-binding domain.

Arthropoda) (Fig. 1C; Table EV1). These ORC6-negative groups span several major metazoan clades, including Deuterostomia, Spiralia, and Ecdysozoa. Our findings suggest that ORC6 has been independently lost in multiple metazoan lineages, highlighting a distinct evolutionary trajectory compared to the strongly conserved ORC1-5 subunits.

## Loss of ORC6 correlates with poor conservation of the ORC6-binding region in ORC3

While our initial analysis supports ORC6 loss in multiple metazoan taxa (Fig. 1C; Table EV1), we sought to rule out alternative explanations, such as incomplete genome sequencing or the presence of highly diverged ORC6 orthologs undetectable by our search methods. To address these possibilities, we examined the

conservation of ORC6's binding partner in ORC, ORC3, focusing on whether the ORC6-BD in the insertion domain of ORC3 is conserved across Metazoa (Fig. 1A,B). Using *C. elegans* as a test case of an ORC6-negative species, we first superposed the structure of human ORC3 (Jaremko et al, 2020) with an AlphaFold model of *C. elegans* ORC3 (Jumper et al, 2021; Tunyasuvunakool et al, 2021). Although the winged helix and AAA+ domains aligned well between the two models, the ORC6-BD and additional parts of the insertion domain were unexpectedly absent in *C. elegans* ORC3 (Figs. 2A and EV1B). To determine whether this correlation extends across Metazoa, we generated a multiple protein sequence alignment (MSA) of ORC3, which revealed strong conservation of residues in the ORC6-binding region among ORC3 orthologs of species with ORC6 (Figs. 2B and EV1C,D); by contrast, ORC3 sequences from organisms without ORC6 had numerous

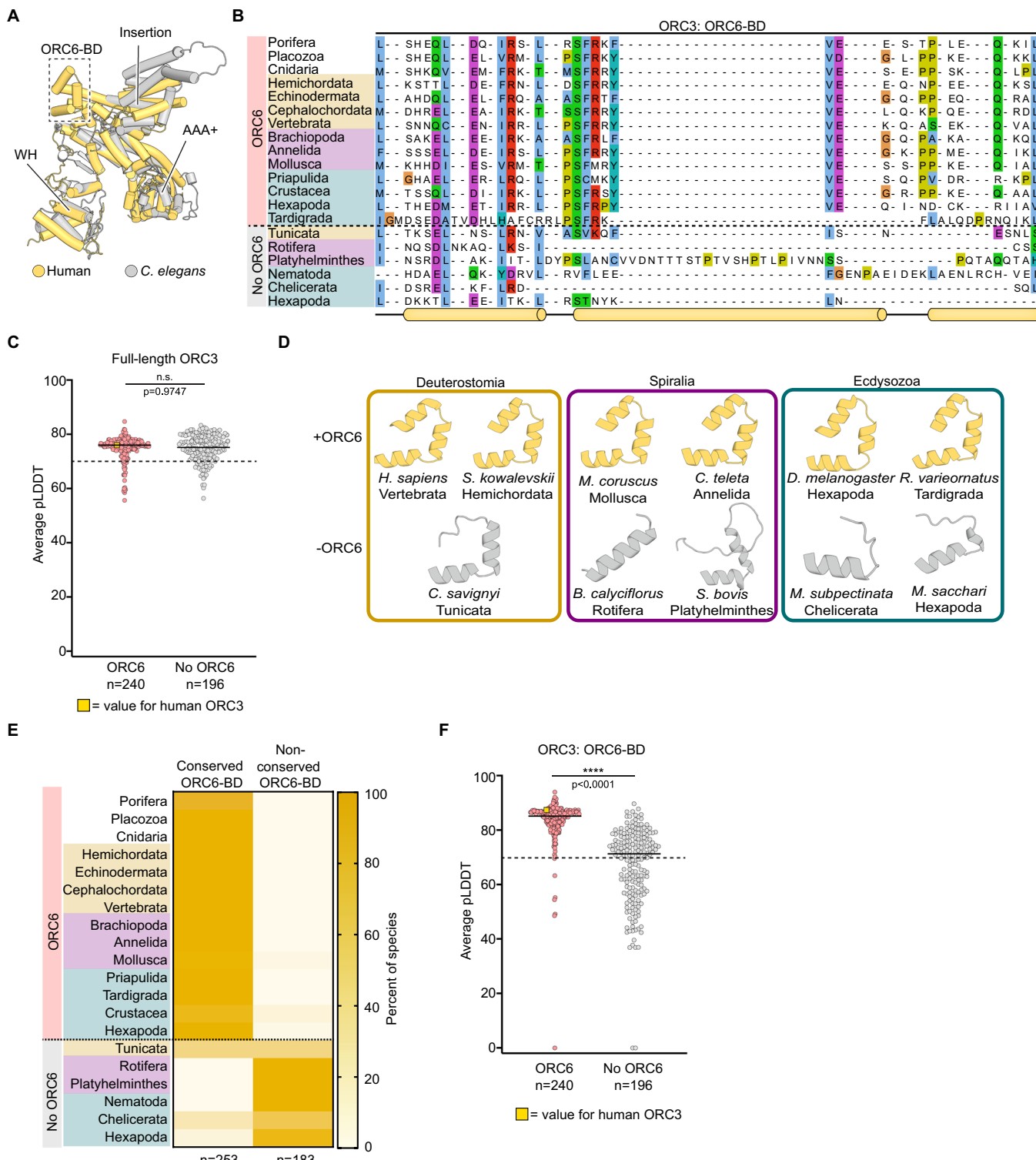

**A** Human, C. elegans — ORC6-BD, Insertion, WH, AAA+

**B** ORC3: ORC6-BD

**C** Full-length ORC3 — Average pLDDT; ORC6 n=240, No ORC6 n=196; n.s. p=0.9747; □ = value for human ORC3

**D** Deuterostomia, Spiralia, Ecdysozoa; +ORC6 / -ORC6
- *H. sapiens* Vertebrata, *S. kowalevskii* Hemichordata, *C. savignyi* Tunicata
- *M. coruscus* Mollusca, *C. teleta* Annelida, *B. calyciflorus* Rotifera, *S. bovis* Platyhelminthes
- *D. melanogaster* Hexapoda, *R. varieornatus* Tardigrada, *M. subpectinata* Chelicerata, *M. sacchari* Hexapoda

**E** Conserved ORC6-BD / Non-conserved ORC6-BD; Percent of species; n=253, n=183

**F** ORC3: ORC6-BD — Average pLDDT; ORC6 n=240, No ORC6 n=196; **** p<0.0001; □ = value for human ORC3

gaps and insertions in this region, suggesting that the ORC6-BD in ORC3 is less conserved in these species (Figs. 2B and EV1C,D).

Surprisingly, we also noticed that a subset of ORC3 sequences belonging to the hexapoda subphylum exhibited large alignment gaps or loss of conserved residues in the ORC6-BD region as compared to other arthropods (Fig. EV1E). We had originally

assumed that ORC6 is conserved across all hexapods, given that a BLAST search for ORC6 orthologs among hexapods yielded over 300 hits. To investigate this observation further, we grouped ORC3 sequences with gaps in the ORC6-BD region based on taxonomy and found that they clustered within discrete hexapod phylogenetic groups: the Collembola class, the Sternorrhyncha

◄ **Figure 2.  The ORC6-binding site in ORC3 is degenerate in species without ORC6.**

(**A**) Structural superposition of human ORC3 (PDB 7JPO (Jaremko et al, 2020)) and *C. elegans* ORC3 (AlphaFold database, accession number Q95Y69). The entire ORC6-BD region is absent in *C. elegans* ORC3. (**B**) Multiple protein sequence alignment of ORC3, highlighting strong sequence conservation of the ORC6-BD in species with ORC6 but poor conservation in species without identified ORC6 orthologs. The location of the three α-helices in the ORC6-BD are marked by cylinders below the alignment. (**C**) Average pLDDT scores of ORC3 ortholog AlphaFold2 predictions. (**D**) Example ORC6-BD folds from AlphaFold2-predicted ORC3 orthologs in ORC6-positive (top, in yellow) and ORC6-negative (bottom, in gray) species. (**E**) Heat map summarizing ORC6-BD conservation across Metazoa in species with and without ORC6, as inferred from AlphaFold2-predicted ORC3 models. Coloring reflects the percentage of species within each taxonomic group with conserved or non-conserved ORC6-BD. The definition of a conserved and non-conserved ORC6-BD can be found in the methods section. (**F**) Average pLDDT scores of the ORC6-BD region from ORC3 ortholog AlphaFold2 predictions. Solid black lines in (**C**, **F**) represent medians, and dotted black lines at pLDDT = 70 mark the accepted threshold for reliable backbone predictions (Tunyasuvunakool et al, 2021). *P* values were calculated to four decimal points using a two-tailed Mann–Whitney test. n.s. no significance. Source data are available online for this figure.

suborder, the Hymenoptera order, and the Lampyridae family. BLAST searches for ORC1-6 within these groups revealed ORC1-5 orthologs, but no ORC6 orthologs, suggesting that these groups have likely lost the ORC6 subunit and that ORC6 conservation is variable among hexapods (Fig. EV1F). For subsequent analyses, we have divided hexapod sequences into two groups based on ORC6 conservation.

Given the poor sequence conservation of ORC3's ORC6-BD in species without ORC6, we next asked whether loss of ORC6 is accompanied by a structural change of its binding site in ORC3 as seen in the *C. elegans* ORC3 AlphaFold model (Figs. 2A,B and EV1B–D). We therefore generated AlphaFold2 (Jumper et al, 2021) models of ORC3 for over 400 metazoan species (Fig. EV2; Dataset EV1). To assess the reliability of our AlphaFold2 predictions, we first confirmed that the domain folds were generally correct by superposing each model with human ORC3, which showed good alignment of the winged helix, AAA+, and insertion domains across all models, similar to Fig. 2A. We then assessed each model using a built-in AlphaFold confidence score, the predicted local distance difference test (pLDDT), a metric for how well the model may agree with an experimentally determined structure (Jumper et al, 2021; Tunyasuvunakool et al, 2021). The majority of ORC3 predictions had an average pLDDT above 70, indicating that the protein backbone for these predictions is generally correct (Tunyasuvunakool et al, 2021) (Fig. 2C; Appendix Fig. S1A). Importantly, there was no significant difference in the average pLDDT score between ORC3 orthologs from species with and without ORC6, confirming that the reliability of the ORC3 models was comparable across both groups (Fig. 2C; Appendix Fig. S1A).

Structural comparison of ORC6-BDs in ORC3 AlphaFold2 models revealed strong conservation in species with but not without ORC6 (Fig. 2D,E). Models of ORC3 from species with ORC6 almost universally retained the characteristic three α-helices of the ORC6-BD (234/240 models or ~98%) that are seen in experimentally determined structures of human and *Drosophila* ORC3 (Bleichert et al, 2015; Jaremko et al, 2020) (Figs. 1B and 2D). By contrast, ~90% (177/196) of ORC3 models from species lacking ORC6 orthologs exhibited structurally degenerate ORC6-BDs, often missing one or more of the α-helices present in the canonical fold (Fig. 2D,E). To estimate the reliability of the ORC6-BD region in these AlphaFold2 models, we analyzed pLDDT scores for residues within the ORC6-BD, using our ORC3 multiple sequence alignments to define the residue range of the ORC6-BD for each species (Fig. 2B). In species with ORC6, the median of the average pLDDT for the ORC6-BD was above 80, indicating this region is of

high confidence in the predicted models and consistent with expectations that ORC3's ORC6-BD is well-conserved and structurally ordered when ORC6 is present in a given species (Fig. 2F; Appendix Fig. S1B). By contrast, the median of the average pLDDT for ORC3's ORC6-BD in organisms without a cognate ORC6 ortholog was barely greater than 70, suggesting lower confidence predictions of this region (Fig. 2F; Appendix Fig. S1B). The lower confidence scores imply that in species without ORC6, the ORC6-BD sequence is degenerate to the extent that AlphaFold cannot confidently identify it as a canonical ORC6-BD. Together, these results provide compelling evidence that the loss of ORC6 across multiple metazoan lineages is accompanied by the degeneration of its binding site in ORC3. Furthermore, they suggest that evolutionary pressure to maintain the ORC6-BD is dependent on the presence of an ORC6 ortholog, which, once lost, allows the binding site to diverge over time.

## Degeneracy of the ORC6-binding domain disrupts in silico ORC3-ORC6 interactions

Our multiple sequence alignments and AlphaFold2 predictions uncovered a remarkable degeneracy of ORC3's ORC6-BD in ORC6-negative species (Figs. 2 and EV2). While it was evident that ORC3 proteins from species that completely lack the ORC6-BD (such as *C. elegans*) would be unable to bind ORC6, we questioned whether other ORC6-BDs (e.g., from tunicates) may be similar enough to the canonical fold to retain ORC6 binding despite slight divergences in fold and sequence (Figs. 2B,D and EV2). Intact ORC6-binding ability among degenerate ORC6-BDs could additionally indicate the presence of undetected ORC6 orthologs. To test this premise, we turned to AlphaFold2 Multimer to predict interactions between ORC6 and ORC3 in silico across the metazoan kingdom. For ORC6-positive species, predictions could be generated using native ORC3-ORC6 pairs, but for ORC6-negative species, this approach was not possible due to the lack of a native ORC6 ortholog. To circumvent this limitation, we used the human ORC6 sequence for all predictions, generating multimer predictions for human ORC6 with all ORC3 sequences in our AlphaFold dataset (Dataset EV2). This approach was justified by the strong sequence conservation of both the ORC6-BD (Fig. 2B) and the ORC6-CTD helix (Bleichert et al, 2013), suggesting that cross-species ORC3-ORC6 pairs likely retain interacting residues.

To evaluate the reliability and accuracy of these predictions, we primarily used a metric called *average models*, which has been previously applied to evaluate small-scale AlphaFold2 Multimer

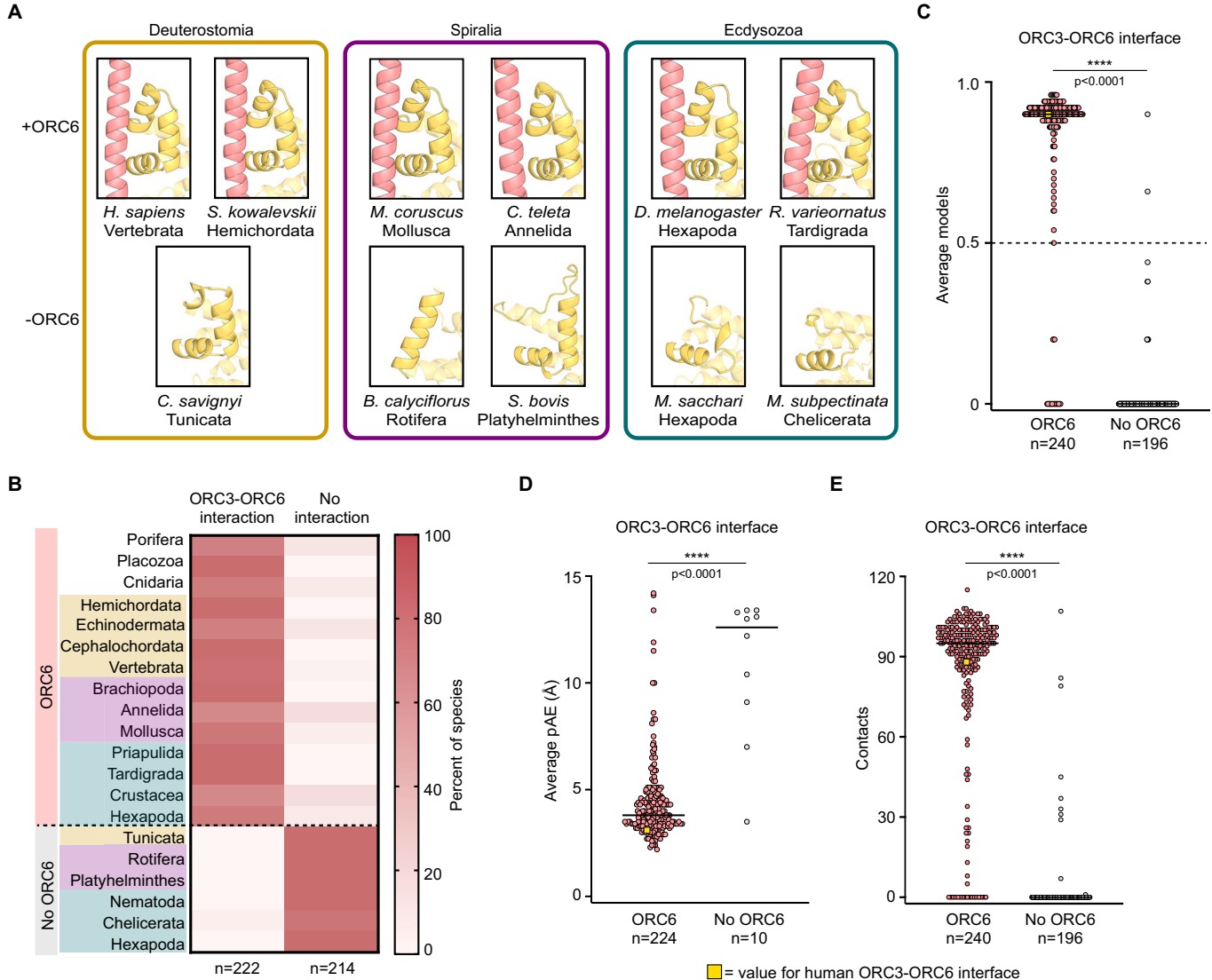

**Figure 3. Degeneracy of the ORC6-binding region in ORC3 leads to loss of ORC6-ORC3 interactions in AlphaFold2 Multimer predictions.**

(A) Examples of AlphaFold2 Multimer predictions using human ORC6 (salmon) and various metazoan ORC3 orthologs (yellow) as input. Canonical ORC3-ORC6 interfaces are predicted between human ORC6 and ORC3 from ORC6-positive species (top) but not between human ORC6 and ORC3 from ORC6-negative species (bottom) despite the inclusion of an ORC6 input sequence. (B) Heat map summarizing ORC3-ORC6 interactions in AlphaFold2 Multimer predictions of human ORC6 and metazoan ORC3 orthologs. An ORC3-ORC6 interaction is defined as an AlphaFold2 Multimer prediction with a canonical ORC3-ORC6 interaction and an *average models* score of > 0.5 as depicted in (C). (C) *Average models* scores for ORC3-ORC6 AlphaFold2 Multimer predictions between metazoan ORC3 orthologs and human ORC6, for species with and without ORC6. *Average models* scores were calculated for the canonical ORC3-ORC6 interface only. Dotted black line at *average models* = 0.5 marks the confidence cut-off. (D) Average interface pAE scores of the canonical ORC3-ORC6 interface from AlphaFold2 Multimer predictions. If no interface was formed, pAE could not be calculated, and no data is shown for these datapoints. (E) Number of contacts between the ORC6-BD and ORC6 in ORC3-ORC6 predictions. Solid black lines in (C–E) represent medians. Values of 0 in (C, E) indicate that no canonical interface was predicted. *P* values were calculated to four decimal points using a two-tailed Mann–Whitney test. Source data are available online for this figure.

protein–protein interaction screens and has been shown to outperform other commonly used metrics such as pDockQ (predicted docking quotient) and ipTM (interface predicted template-modeling) scores (preprint: Evans et al, 2021; Bryant et al, 2022; Lim et al, 2023; Schmid and Walter, 2025). The *average models* score quantifies how consistently AlphaFold's five output models predict the same interaction interface and was calculated in a similar manner as described in Lim et al, 2023 and Schmid and Walter, 2025 (see also Methods). This score reflects the proportion

of contacts shared across all five models, with a score of 1 indicating perfect agreement among models, while a score of 0 suggests either no interactions, completely inconsistent binding predictions, or low-confidence contacts. For analysis of ORC3-ORC6 predictions, we limited *average models* scoring to contacts involving ORC3 residues within the ORC6-BD, reflecting the canonical, experimentally validated interaction between the ORC6-BD in ORC3 and the ORC6-CTD (Bleichert et al, 2013; Bleichert et al, 2015), and excluding any predicted non-canonical interactions. An *average*

*models* score threshold of 0.5 was used as the cutoff for defining a prediction as a confident ORC3-ORC6 interaction as described in Schmid and Walter, 2025.

AlphaFold2 Multimer consistently predicted confident (defined by *average models* > 0.5), canonical interactions between human ORC6 and ORC3 for 220 out of 240 ORC6-positive species (~92%), even those distantly related to humans (Figs. 3A,B and EV3). The median *average models* score for these predictions was 0.9, indicating strong agreement across all five models for most predictions (Fig. 3C; Appendix Fig. S2A). This value matches the score for the native human ORC3-ORC6 complex (0.9) and, notably, also aligns with the score for a native *D. melanogaster* ORC3-ORC6 model (0.94), both interactions that have been experimentally validated (Bleichert et al, 2013; Bleichert et al, 2015). Among predictions between human ORC6 and ORC3 from ORC6-negative species, only 10/196 (5%) had *average models* scores greater than 0 (Fig. 3C; Appendix Fig. S2A). Of these ten, only two correspond to a canonical ORC3-ORC6 interaction. Both are from chelicerate species with canonical ORC6-BDs and may represent organisms that have lost ORC6 more recently.

The classification based on *average models* scores was supported by additional AlphaFold2 Multimer metrics, including interface pAE (the average predicted aligned error among residue pairs at the canonical ORC3-ORC6 interface) and ipTM (the interface predicted template-modeling) scores (preprint: Evans et al, 2021). For ORC6-positive species, the median interface pAE was 3.8 Å and the median ipTM was 0.56, closely matching the values for native human (3.1 Å, 0.61) and *D. melanogaster* (3.4 Å, 0.66) complexes (Fig. 3D; Appendix Fig. S2B,C). Similarly, the number of predicted interfacial contacts (median = 95) was comparable to that of the human ORC3-ORC6 model (88 contacts) and the native *D. melanogaster* model (133 contacts) (Fig. 3E; Appendix Fig. S2D). For ORC6-negative species, the median number of contacts was 0, the median interface pAE was 12.6 Å, and the median ipTM was 0.2, reflecting the fact that very few ORC3-ORC6 interfaces were predicted for these species (Fig. 3D,E; Appendix Fig. S2B–D). Collectively, these results indicate that the non-conserved ORC6-BDs in ORC6-negative species are likely nonfunctional for ORC6 binding, providing further support that both ORC6 and the ORC6-BD in ORC3 have been lost in many metazoan lineages.

## The ORC3 tether is conserved across Metazoa for ORC6-independent MCM loading

ORC6-positive species can license origins in an ORC6-dependent manner through an MO-I intermediate that relies on ORC6 to position ORC in the opposite orientation for the second MCM loading event (Yang et al, 2024; Weissmann et al, 2024). An alternative MO-II complex has been proposed to perform a similar function in an ORC6-independent manner, requiring an interaction between ORC3 and MCM (Yang et al, 2024). We therefore examined whether the ORC3 element mediating these contacts and ORC3-MCM interactions are conserved across Metazoa.

MO-II complex assembly requires the ORC3 tether, a loop-helix motif located in the insertion domain of ORC3 (Fig. 1A) that binds the N-terminal domain of MCM2 (Fig. 4A) (Yang et al, 2024). Analysis of our metazoan ORC3 AlphaFold2 predictions revealed that the ORC3 tether is more ubiquitous across the kingdom than ORC6 (Fig. 4B). Only 18 ORC3 orthologs, including most orthologs

from Rotifera and all from Tardigrada, lacked either a loop-helix fold in this region or a linker of sufficient length to form one (Figs. 4B and EV2). The ORC3 tether is intrinsically flexible, as evidenced by its absence from high-resolution metazoan ORC structures (Bleichert et al, 2015; Schmidt and Bleichert, 2020; Jaremko et al, 2020; Weissmann et al, 2024). Consistent with its flexible nature, the pLDDT scores for the ORC3 tether region (median = 54.67, Fig. EV4A) are lower than those for the more ordered ORC6-BD (median = 85.11, Fig. 2F; Appendix Fig. S1B) but similar to the average pLDDT of the human ORC3 tether (49.30). Multiple protein sequence alignments of ORC3 showed some conserved hydrophobic residues in the ORC3 tether helix and a conserved lysine residue adjacent to the helix; sequences from the Rotifera and Tardigrada orthologs, on the other hand, contain gaps in the helix region and are missing the conserved lysine (Fig. EV4B). Collectively, these results support the notion that a flexible ORC3 tether element is commonly found across Metazoa, opening the possibility that an ORC3 tether may facilitate origin licensing in metazoans, including non-human species.

To better understand the conservation of the ORC3 tether and whether all identified ORC3 tethers across Metazoa could interact with MCM2, we compiled MCM2 orthologs from species corresponding to our ORC3 dataset and generated AlphaFold2 Multimer predictions of native ORC3-MCM2 complexes (Fig. EV5A–C; Dataset EV3). Many ORC3-MCM2 pairs formed similar contacts to those observed in the human ORC3-MCM2 complex, exemplified by a comparison of the human and *C. elegans* predicted models (Figs. 4A and EV5A–C). As in our ORC3-ORC6 AlphaFold2 Multimer analysis, we calculated *average models* scores for these predictions, considering only contacts involving the ORC3 tether, and set a confidence threshold of *average models* score > 0.5. Due to the flexible nature of the ORC3 tether and the small size of the interface, we also compared AlphaFold2 Multimer confidence metrics to the human ORC3-MCM2 model as a benchmark for a functionally important interaction that has been experimentally validated (Yang et al, 2024). AlphaFold2 Multimer predicted confident ORC3-MCM2 interactions for 354 out of 384 (92%) species with an ORC3 tether (Fig. 4C) with a median *average models* score across all groups of 0.72, in agreement with the human ORC3-MCM2 average models score of 0.76 (Figs. 4D left and EV5D). The median interface pAE (8.2 Å) was also similar to the value for the human ORC3-MCM2 prediction (interface pAE = 8.1 Å, Figs. 4E and EV5E). The only group with orthologs containing ORC3 tethers and consistently low-confidence predictions (average models score < 0.5) was Porifera (Fig. EV5D), which may suggest that conservation of the ORC3 tether is variable in this phylum. ORC3 proteins lacking sufficient residues in the ORC3 tether region to form a loop-helix fold, although a small sample size, failed to form any consistent contacts with MCM2 as evidenced by average models scores of 0 (interface pAE values could not be calculated due to the lack of predicted interfaces; Figs. 4D left,E, and EV5D,E). Notably, *average models* scores did not differ between species with and without ORC6 (Fig. 4D right), indicating that ORC3 tether conservation is independent of ORC6's presence. Overall, these results indicate that the ORC3 tether region and ORC3-MCM2 interactions are widely conserved across Metazoa (Fig. 4F) and provide a mechanistic explanation for how ORC6-negative species could efficiently license DNA replication origins despite the absence of ORC6.

 

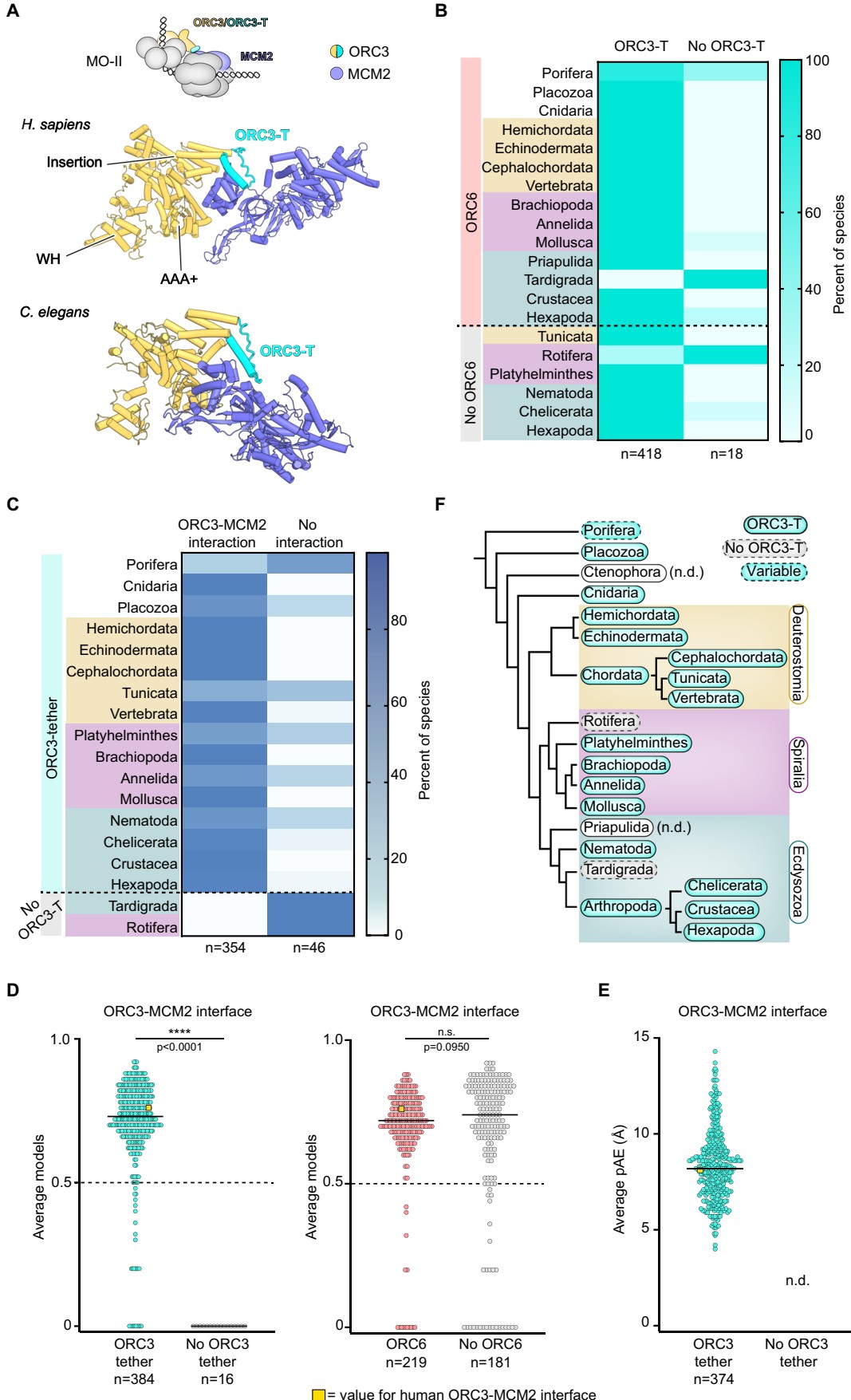

◄ **Figure 4.  The ORC3 tether is conserved across most metazoan species.**

(A) The ORC3 tether binds to MCM2 to stabilize ORC at the N-terminal face of the first loaded MCM hexamer in the MO-II intermediate (cartoon). AlphaFold2 Multimer predictions for interactions between ORC3 and MCM2 for human and *C. elegans* are shown below. (B) Heat map summarizing the presence of an ORC3 tether across Metazoa, based on AlphaFold2 predictions of ORC3 orthologs. Coloring reflects the percent of species where an ORC3 tether could or could not be identified within each taxonomic group. (C) AlphaFold2 Multimer predictions reveal conservation of ORC3-MCM2 binding in taxonomic groups with the ORC3 tether. A heat map summarizing the presence of ORC3-MCM2 binding interfaces in AlphaFold2 Multimer predictions of ORC3-MCM2 complexes across Metazoa is shown. An interaction is defined as an *average models* score of > 0.5 as depicted in (D). (D) The ORC3-MCM2 interaction is equally conserved across species with and without ORC6 orthologs but absent in species missing an ORC3 tether. *Average models* scores for AlphaFold2 Multimer ORC3-MCM2 interactions for species with and without an ORC3 tether (left) and for species with and without an ORC6 ortholog (right) are plotted. *Average models* scores are based on contacts within the canonical ORC3-MCM2 interface. Dotted black lines at *average models* = 0.5 mark the confidence cut-off. (E) Average interface pAE scores for the ORC3 tether-MCM2 interface in AlphaFold2 Multimer predictions. If no interface was formed, pAE could not be calculated, and no data are shown for these datapoints. Solid black lines in (D, E) represent medians. Values of 0 in (D) indicate that no canonical ORC3-MCM2 interface was predicted. *P* values were calculated to four decimal points using a two-tailed Mann–Whitney test. (F) Phylogenetic tree summarizing functional ORC3 tether conservation among metazoan taxonomic groups. Tree construction was based on widely accepted metazoan phylogeny (Laumer et al, 2019) and distances of branches depicted are arbitrary. ORC3-T ORC3 tether, n.d. no data available, n.s. no significance. Source data are available online for this figure.

## Discussion

Redundant and diverse molecular mechanisms are common in biology, particularly in essential processes, where they help ensure fidelity and robustness under variable cellular conditions. For example, multiple layers of redundancies ensure that origin licensing is restricted to the G1 phase of the cell cycle to prevent relicensing and re-replication of newly synthesized DNA. In yeast, phosphorylation of ORC subunits, Cdc6, and MCM2-7 all prevent MCM loading in an overlapping manner (Diffley, 2010). Here, we explore the evolutionary conservation and physiological relevance of multiple pathways through which the second copy of the metazoan MCM2-7 hexamer is recruited and loaded onto replication origins to establish the MCM double hexamer as a precursor to bidirectional replication, using a novel structural phylogenetics approach. Recent biochemical and structural experiments using human proteins have suggested three distinct routes for forming MCM double hexamers (Yang et al, 2024; Weissmann et al, 2024) (Fig. 5), a surprising deviation from yeast MCM loading. However, the individual functional importance of each of these human pathways in the cellular environment could not be resolved from in vitro biochemical assays.

The evidence presented here argues for an important physiological role of ORC6-independent origin licensing mechanisms across Metazoa. The loss of ORC6 has occurred independently in multiple metazoan lineages, suggesting that it is an example of convergent secondary gene loss, and that ORC6-independent MCM loading is an ancestral feature rather than an adaptation that evolved exclusively in ORC6-negative species (Porter and Crandall, 2003; Aves et al, 2012; Kato et al, 2024). We propose that all three human MCM loading mechanisms are broadly conserved across Metazoa, with different lineages specializing in particular pathways. For example, species retaining the ORC3 tether but not ORC6, including tunicates, flatworms, nematodes, chelicerates, and some hexapods, may have adapted to primarily utilize the MO-II-mediated pathway for MCM loading (Fig. 5, **mechanism 2**). Conversely, species with an ORC6 ortholog but without the ORC3 tether may have specialized in the MO-I-mediated, ORC6-dependent pathway (Fig. 5, **mechanism 1**). The MO-independent pathway (Fig. 5, **mechanism 3**) is likely common to all Metazoa as it relies neither on ORC6 nor the ORC3 tether. The ability to load the MCM helicase through multiple pathways may confer advantages to multicellular eukaryotes in particular, which must

execute developmental programs requiring rapid cellular division during early life stages.

Among all groups included in our data, rotifers are particularly intriguing, as they have lost both the ORC6 subunit and the ORC3 tether, rendering them unable to establish an MO complex. In the MO-independent MCM loading pathway, the correct orientation of the second MCM hexamer being recruited for double hexamer formation is determined by chance, a 1 in 4 probability, suggesting that relying solely on this mechanism to license an entire genome would be remarkably inefficient. This characteristic raises the possibility that rotifers, and perhaps other metazoan species, possess alternative mechanisms to direct MCM loading orientation. Chromatin context, transcription activity, and other genome features including AT-rich sequences, DNA geometry, and G4 motifs have been proposed to regulate origin licensing in Metazoa (Prioleau and MacAlpine, 2016; Parker et al, 2017; Ekundayo and Bleichert, 2019; Ahmad et al, 2024) and may potentially play a role in this regard, but such mechanisms are still poorly understood.

The reason why ORC6 has been lost in certain metazoan taxa but not others remains unclear, but may represent an evolutionary adaptation to distinct regulatory requirements and genome organization. We note that we cannot definitively rule out that some species categorized here as ORC6-negative could possess a highly divergent ORC6 ortholog that is undetectable by BLAST searches; however, our search methods were sufficiently sensitive to return *S. cerevisiae* Orc6 with human ORC6 as a query, which is notable as *S. cerevisiae* Orc6 contains a 200-amino acid linker unique to yeasts. The identified ORC6-negative species are enriched for parasites, including all flatworm species and many nematode species in our dataset, where genome streamlining and gene loss are common (Porter and Crandall, 2003; Wolf and Koonin, 2013; Poulin and Randhawa, 2015). However, we also identified a significant number of free-living ORC6-negative species, and not all parasitic metazoans are missing an ORC6 subunit. Likewise, genome size varies widely in both ORC6-positive and ORC6-negative groups (Gregory, 2004; Hotaling et al, 2021). While genome size does not necessarily correlate with organismal complexity, it generally scales with the number of origins that must be licensed for successful replication (Prioleau and MacAlpine, 2016; Parker et al, 2017). Notably, ORC6 is an essential protein in humans, other vertebrates, and *Drosophila* (Prasanth et al, 2002; Balasov et al, 2007; Bernal and Venkitaraman, 2011), although this property may be attributed not only to ORC6's role in

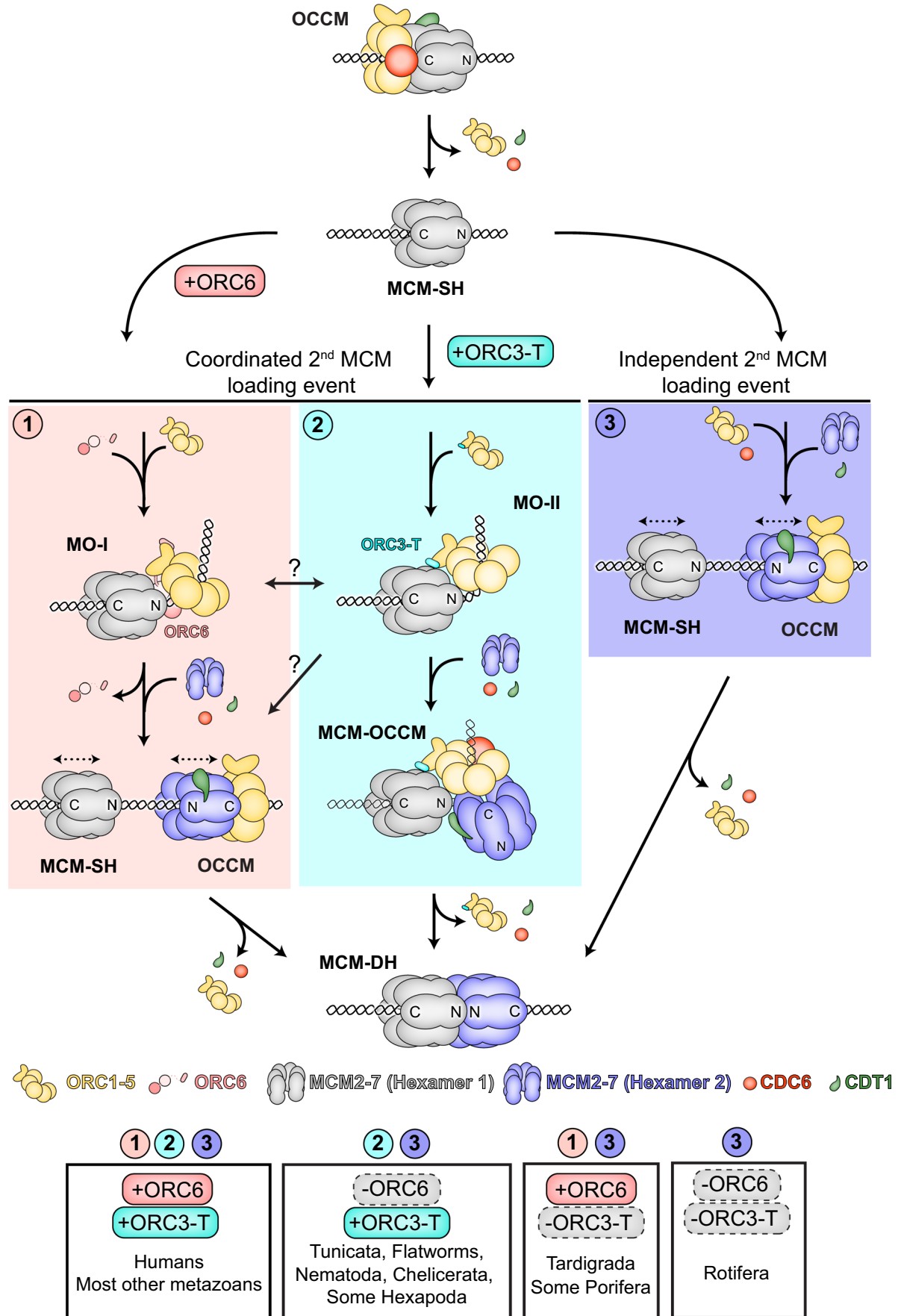

◀ **Figure 5. Model of origin licensing pathways across Metazoa.**

Model depicts proposed MCM-DH loading pathways based on structural and biochemical experiments of human origin licensing (Weissmann et al, 2024; Yang et al, 2024). The first MCM-SH is loaded through an OCCM intermediate. The second MCM-SH could be loaded through three distinct mechanisms. (1) The MO-I-dependent pathway requires ORC6; the MO-I complex orients ORC for the second MCM-SH loading event, which would occur uncoupled from the MO-I, followed by self-association of the MCM-SHs to form an MCM-DH. (2) The MO-II-dependent pathway requires the ORC3 tether; the MO-II complex could directly recruit a second MCM-SH to load the MCM-DH or, in the presence of ORC6, may shift into the MO-I conformation. (3) The second MCM-SH is loaded independently of the first, with the correct head-to-head orientation arising by chance. The two independently loaded MCM-SH self-associate to form the MCM-DH. Species with a conserved ORC6 ortholog and ORC3 tether would support all three loading pathways. Species without an ORC6 ortholog but with an ORC3 tether could use pathways (2) and (3). Species with an ORC6 ortholog but not an ORC3 tether could load MCM-DH through pathways (1) and (3). Species without ORC6 or the ORC3 tether could only load MCM-DH through pathway (3).

origin licensing but also to several non-replicative functions, for example in cytokinesis (Prasanth et al, 2002; Chesnokov et al, 2003; Sasaki and Gilbert, 2007; Huijbregts et al, 2009; Bernal and Venkitaraman, 2011). These multifunctional roles may have evolved in certain metazoan taxa, while ORC6-negative species may have repurposed other cellular machinery to fulfill these functions. Further investigations are needed to determine how these species compensate for ORC6 loss and whether they have adapted unique mechanisms to maintain genome stability.

Our findings add to the growing body of evidence suggesting that mechanisms for loading the replicative helicase onto DNA are more heterogenous across eukaryotes than previously thought (Godoy et al, 2009; Aves et al, 2012; Tiengwe et al, 2012; Marques et al, 2016; Salas-Leiva et al, 2021). All eukaryotes possess replication machineries with an evolutionary lineage that can be traced to Archaea (Leipe et al, 1999; Makarova and Koonin, 2013). Accordingly, most eukaryotes, including yeast and Metazoa, share a conserved core of origin licensing proteins comprised of ORC1-5, CDC6, CDT1, and MCM2-7. However, lineage-specific insertions in core ORC subunits can impart distinct functions, such as the Orc4 insertion helix in *S. cerevisiae*, which confers sequence-specific origin recognition unique to budding yeasts (Hu et al, 2020; Lee et al, 2021). Similarly, the loss of ORC6 in some Metazoa parallels other examples of missing core origin licensing components among eukaryotes. Some eukaryotic groups, including trypanosomes and metamonads, have undergone extensive loss of canonical genome maintenance proteins like CDT1 and multiple ORC subunits (Godoy et al, 2009; Aves et al, 2012; Tiengwe et al, 2012; Marques et al, 2016; Salas-Leiva et al, 2021). While many of these organisms are parasitic, the recent discovery of a free-living protist lacking canonical ORC orthologs suggests that loss of canonical genome maintenance proteins is not limited to parasites (Salas-Leiva et al, 2021). These findings underscore the flexibility of eukaryotic replication systems and the potential for alternative origin licensing strategies.

Beyond insights into origin licensing mechanisms across Metazoa, our study also underscores the power of AlphaFold in tackling diverse biological problems. Since its introduction, studies employing unique applications of AlphaFold have proliferated. For instance, researchers have leveraged AlphaFold-based structural clustering to study protein evolution (Barrio-Hernandez et al, 2023), identified DONSON as a crucial missing link in the establishment of bidirectional replication through a protein–protein interaction screen (Lim et al, 2023), and integrated AlphaFold predictions with cryo-EM to achieve an unprecedented, near-complete model of the nuclear pore complex (Mosalaganti et al, 2022). Building on this foundation, we employed a uniquely creative AlphaFold-driven approach to address a question inaccessible to traditional methods used to study replication origin licensing, i.e., in vitro reconstitution and cryo-EM structure determination. While these techniques excel at resolving molecular mechanisms in a specific isolated system, they are impractical for systematically analyzing protein–protein interactions across an entire kingdom of life. By leveraging AlphaFold to predict the conservation and structural variability of ORC subunits and their interactions, we developed a framework that enhances traditional phylogenetic methods such as multiple sequence alignment, which offers valuable insight but lacks tertiary and quaternary structure information.

It is important to note that our study is limited by the availability of animal genomic data, which represents only a fraction of total animal diversity, leading to potential biases. Certain taxonomic groups are overrepresented or underrepresented in sequencing databases, which may affect the generalizability of our conclusions. Additionally, while AlphaFold can achieve near-experimental accuracy in the prediction of numerous protein targets (preprint: Evans et al, 2021; Jumper et al, 2021; Tunyasuvunakool et al, 2021; Abramson et al, 2024), it still has several blind spots, particularly with respect to accurately predicting conformational variability and intrinsically disordered regions, as well as gaps in accuracy for specific protein families, such as antibodies or orphan proteins (Yang et al, 2023; Yin and Pierce, 2024; Lopez-Sagaseta and Urdiciain, 2025; Malhotra et al, 2025). While AlphaFold confidence metrics help evaluate predictions, it remains essential to interpret results in conjunction with experimental data (Terwilliger et al, 2024). To mitigate these limitations, we have analyzed ORC3, ORC3-ORC6, and ORC3-MCM2 predictions in the context of experimentally solved structures and previous experimental validations of protein–protein interactions in our complexes (Bleichert et al, 2013; Bleichert et al, 2015; Jaremko et al, 2020; Schmidt and Bleichert, 2020; Yang et al, 2024; Weissmann et al, 2024).

To directly confirm the MCM loading mechanisms proposed here, future studies should focus on in vitro reconstitution of origin licensing for metazoan systems with divergent licensing machinery lacking ORC6 and/or the ORC3 tether. Experimental validation of origin licensing mechanisms in organisms that naturally lack an ORC6 subunit will enable further insights into how both ORC6-dependent and ORC6-independent MCM loading mechanisms contribute to replicative helicase loading and maintenance of genome stability in multicellular eukaryotes. These investigations will further clarify the evolutionary pressures that have shaped MCM loading mechanisms and their broader implications for DNA replication across eukaryotes.

# Methods

### Reagents and tools table

| Reagent/resource | Reference or source | Identifier or catalog number |
|---|---|---|
| **Experimental models** | | |
| N/A | | |
| **Recombinant DNA** | | |
| N/A | | |
| **Antibodies** | | |
| N/A | | |
| **Oligonucleotides and other sequence-based reagents** | | |
| N/A | | |
| **Chemicals, Enzymes and other reagents** | | |
| N/A | | |
| **Software** | | |
| AlphaFold v2.3.2 | Jumper et al, 2021 | |
| colabfold_analysis.py | Lim et al, 2023 https://zenodo.org/records/8223143 | |
| blastp, DELTA-BLAST | Altschul et al, 1997 Boratyn et al, 2012 https://blast.ncbi.nlm.nih.gov/Blast.cgi | |
| MAFFT version 7 | Katoh et al, 2019 https://mafft.cbrc.jp/alignment/server/index.html | |
| PyMOL version 3.0.0 | PyMOL Molecular Graphics System, Schrödinger, LLC | |
| GraphPad Prism 10 | www.graphpad.com | |
| **Other** | | |
| N/A | | |

## ORC1-6 ortholog search

Orthologs for all ORC subunits were identified using blastp (Altschul et al, 1997) and DELTA-BLAST (Boratyn et al, 2012) searches against the NCBI non-redundant protein sequences database with the human full-length sequences as search queries (UniProt accession numbers: ORC1 – Q13415, ORC2 – Q13416, ORC3 – Q9UBD5, ORC4 – O43929, ORC5 – O43913, ORC6 – Q9Y5N6). Targeted searches were first carried out for each metazoan phylum represented by protein entries in NCBI. Phyla were excluded if orthologs for more than one ORC subunit were entirely absent across all species with available sequence data in that phylum. Additional subphylum-level searches were performed within the Chordata and Arthropoda phyla. DELTA-BLAST was used for increased search sensitivity when BLASTP returned limited or no hits. The DELTA-BLAST search method is more sensitive than blastp, being capable of detecting homology between human and budding yeast ORC6, whereas blastp fails to do so. A secondary search of the UniProtKB database was carried out for phyla/subphyla for which less than ten species were identified from NCBI. Results were curated to select one sequence per ORC subunit per species; if multiple sequences were identified, the sequence with the highest query coverage was retained as the representative sequence. False positives, including misassigned ORC subunits and homologous proteins such as CDC6, were distinguished based on percent identity scores, which varied by subunit and taxonomic group. For example, within Mollusca, ORC1 homologs generally had a percent identity score between 55 and 65%, while false-positive CDC6 homologs had percent identity scores of <30%.

## MCM2 ortholog search

Orthologs for MCM2 were identified using targeted blastp searches for each phylum and subphylum included in the analysis, with the full-length human MCM2 sequence as a search query (Uniprot accession number P49736). MCM2 orthologs belonging to species represented in the ORC3 AlphaFold dataset were curated from the results. When multiple sequences per species were identified, the sequence with the highest query coverage was retained as the representative sequence. MCM2 orthologs were differentiated from other MCM subunits by the percent identity score, which was generally >50% for MCM2 sequences.

## Multiple sequence alignment of ORC3

The multiple sequence alignment for ORC3 was generated from 521 diverse metazoan ORC3 sequences, which included all ORC3 sequences in the ORC3 AlphaFold dataset, by MAFFT (Katoh and Standley, 2013; Katoh et al, 2019) using the E-INS-i alignment method.

## Percent identity calculation

Pairwise sequence percent identity scores between metazoan ORC6-BDs and the human ORC6-BD were derived from the ORC3 multiple sequence alignment. The percent identity score was calculated as the number of matches between a metazoan ORC6-BD and the human ORC6-BD divided by the number of positions in the reference (the human ORC6-BD), excluding gaps, and multiplied by 100 (Thompson et al, 1994).

## AlphaFold2 predictions

AlphaFold monomer predictions of ORC3 were carried out on Yale's McCleary HPC cluster, which provides access to multi-GPU nodes. Predictions were generated by AlphaFold v2.3.2 (Jumper et al, 2021) using max_template_date=2021-11-01, the monomer_ptm model, and no AMBER relaxation. Multimer predictions of cross-species ORC3-ORC6 interactions (metazoan ORC3 orthologs and human ORC6, Uniprot accession number Q9Y5N6) (Dataset EV2) and native ORC3-MCM2 pairs (Dataset EV3) were generated using the multimer mode in AlphaFold v2.3.2 (preprint: Evans et al, 2021), max_template_date=2021-11-01, and no AMBER relaxation. Five total predictions were generated for each target. A model of the native *D. melanogaster* ORC3-ORC6 pair (*Dm*ORC3: NCBI accession number AAD39472.1, *Dm*ORC6 Uniprot accession number Q9Y1B2) was generated by the same methods for comparison against the dataset folded with human ORC6.

## Sample size selection for AlphaFold predictions

The sample size for AlphaFold prediction datasets was largely determined by the availability of full-length ORC3 sequences. For taxonomic groups with <100 ORC3 sequences identified in our ortholog search, AlphaFold predictions for all full-length ORC3 sequences were generated. Ctenophores were excluded from AlphaFold analysis, as sequencing data were available for only one species and no ORC3 ortholog could be identified (although ORC1, ORC2, ORC4, ORC5, and ORC6 orthologs were present). Partial ORC3 sequences were included only for taxonomic groups with very limited data ($n < 10$) and if the sequence truncation was not within the insertion domain (where both the ORC6-BD and ORC3 tether are located), corresponding to five partial ORC3 sequences from the following species: *Hypsibius exemplaris* (Tardigrada), *Amphimedon queenslandica* (Porifera), *Trichoplax adhaerens* (Placozoa), *Capitella teleta* (Annelida), and *Dimorphilius gyrociliatus* (Annelida). Low-quality ORC3 sequences, defined as those with undetermined residues denoted as "X" in the protein sequence, were also excluded.

To use computational resources efficiently and to balance sample sizes across taxonomic groups, a subset of representative sequences for vertebrates and hexapods, for which >100 full-length ORC3 sequences were identified, were chosen as follows. Hexapods were first split into two groups (no ORC6 and +ORC6) according to conservation of ORC6 (Fig. EV1F). All full-length ORC3 sequences from hexapods in groups without ORC6 orthologs ($n = 60$) were included. Among hexapods with ORC6 orthologs, 283 total ORC3 sequences were retrieved, spanning 13 unique orders. From these, 44 representative full-length ORC3 sequences were selected for inclusion in the AlphaFold dataset, prioritizing selection of full-length ORC3 sequences and order-level taxonomic diversity. For vertebrates, over 1000 ORC3 sequences were identified. From these, 64 full-length representative sequences were selected, prioritizing class and order-level taxonomic diversity. The human sequence was added to this group, for a final $n$ of 65. This group contains sequences spanning 11 unique classes and 65 unique orders.

All ORC3 sequences ($n = 436$) included in the ORC3 AlphaFold2 monomer dataset were also used in the ORC3-ORC6 AlphaFold2 Multimer dataset. For the ORC3-MCM2 multimer dataset, all ORC3 sequences from the ORC3 monomer dataset for which a corresponding MCM2 sequence could be found were included ($n = 400$). These MCM2 sequences were both full-length and partial. Partial MCM2 sequences were excluded only if the N-terminal domain that interacts with the ORC3 tether was missing. Low-quality MCM2 sequences, defined as those with undetermined residues denoted as "X" in the protein sequence, were excluded.

## Identification of conserved or non-conserved ORC6 binding domains and ORC3 tethers in ORC3 AlphaFold models

Manual inspection of ORC6-BD conservation in ORC3 was carried out by structural superposition of ORC3 AlphaFold predictions to an experimentally determined human ORC3 structure (PDB 7JPO (Jaremko et al, 2020)) in PyMOL (PyMOL Molecular Graphics System, Schrödinger, LLC) and subsequent comparison of the protein backbone in the ORC6-BD region. Conserved ORC6-BDs

were defined as containing all three α-helices that are observed in the canonical ORC6-BD motif, for example, in human (Jaremko et al, 2020) or *Drosophila* ORC3 (Bleichert et al, 2015). Non-conserved ORC6-BDs were defined as those missing one or more of the three α-helices in the canonical ORC6-BD, as well as those containing insertions within this region. Conservation of the ORC3 tether was assessed in a similar manner. Due to the flexibility of this region and low-confidence (pLDDT) scores, all ORC3 proteins with sufficient residues to form the loop-helix element in the ORC3 insert were defined as containing a likely ORC3 tether, regardless of whether AlphaFold predicted an α-helix in this region.

## Confidence analysis of AlphaFold predictions

Contacts between proteins in pairwise AlphaFold2 Multimer predictions and the *average models* score were calculated using modified python scripts published previously (Lim et al, 2023). Potential contacts between residues were first identified by non-hydrogen atoms <8 Å apart. These contacts were then filtered by the following parameters. To exclude very low-confidence regions, any contacts involving residues with a predicted local distance difference test (pLDDT) value of <50, and any residue pairs with a predicted aligned error (pAE) of >15 Å were excluded. To specifically assess canonical interactions between the ORC6-BD and ORC6 in ORC3-ORC6 predictions, we further filtered out any contact pairs that did not involve a residue in the ORC6-BD of ORC3. For this step, the residue range for the ORC6-BD region in each ORC3 sequence was determined using the ORC3 multiple protein sequence alignment (MSA). Likewise, to specifically assess canonical interactions between the ORC3 tether region and MCM2 in ORC3-MCM2 predictions, the contacts were filtered to include only those involving an ORC3-tether residue, as determined by the ORC3 MSA. Average interface pAEs were calculated by averaging the pAE values for all contact pairs identified within the canonical interfaces. When no interface contacts were predicted, the average pAE could not be calculated. The *average models* score was calculated as the proportion of identified contacts that are shared across each of the five predicted models in the same manner as "average $n$ models" in references (Lim et al, 2023; Schmid and Walter, 2025) and normalized to 1. For ORC3-ORC6 predictions, the *average models* calculation only considered contacts involving the ORC6-BD in ORC3. For the ORC3-MCM2 predictions, *average models* only considered contacts involving the ORC3 tether. Open AI's GPT-4 was used to assist with coding tasks.

## Statistical analysis

$P$ values reported in figures were calculated in GraphPad Prism using a two-tailed Mann–Whitney test since the data did not follow normal distribution. Normality was assessed using the Shapiro-Wilk test.

## Phylogenetic tree generation

Phylogenetic relationships depicted in Figs. 1C and 4F were based on broadly accepted metazoan phylogeny as reported in reference (Laumer et al, 2019). Phylogenetic relationships between hexapod orders as depicted in Fig. EV1F were based on reference (Misof et al, 2014).

## Structure analysis

Structures were visualized and figures rendered using PyMOL (PyMOL Molecular Graphics System, Schrödinger, LLC).

# Data availability

Source data is available online, accompanying the manuscript. AlphaFold2-predicted models generated from this study are available from the authors upon reasonable request.

The source data of this paper are collected in the following database record: biostudies:S-SCDT-10_1038-S44318-025-00628-5.

# Peer review information

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

## Acknowledgements

We thank the Yale Center for Research Computing for guidance and use of the McCleary HPC Cluster. We thank Yong Xiong for helpful feedback on the interpretation of AlphaFold confidence scores. This work was supported by the National Institutes of General Medicine (R01-GM141313 and R35-GM158287 to FB) and the National Cancer Institute (1F31-CA278331 to OH). FB is a member of the Yale Cancer Center.

## Author contributions

**Olivia Hunker**: Data curation; Formal analysis; Funding acquisition; Investigation; Visualization; Methodology; Writing—original draft; Writing—review and editing. **Franziska Bleichert**: Conceptualization; Formal analysis; Supervision; Funding acquisition; Visualization; Writing—review and editing.

Source data underlying figure panels in this paper may have individual authorship assigned. Where available, figure panel/source data authorship is listed in the following database record: biostudies:S-SCDT-10_1038-S44318-025-00628-5.

## Disclosure and competing interests statement

The authors declare no competing interests.

# Expanded View Figures

**Figure EV1.  ORC6 conservation is variable across Metazoa and within the *Hexapoda* subphylum.**                                          ▶

(**A**) Overview of proposed human origin licensing pathways. (**B**) Comparison of human ORC3 (PDB 7JPO (Jaremko et al, 2020)) and *C. elegans* ORC3 (AlphaFold database, accession number Q95Y69), demonstrating loss of the entire ORC6-BD region in *C. elegans* ORC3. (**C, D**) Percent sequence identity of metazoan ORC6-BDs as compared to the human ORC6-BD. Species without an ORC6 ortholog have greater sequence-level variation from the human ORC6-BD than species with an ORC6 ortholog. A comparison of percent identities for species with and without an ORC6 ortholog (in **C**) and for each taxonomic group surveyed (in **D**) is shown. Solid black lines indicate median values. (**E**) Multiple ORC3 protein sequence alignment shows variable conservation of the ORC6-BD in several hexapod orders. (**F**) Phylogenetic tree summarizing the presence or absence of ORC6 orthologs across hexapoda. All groups have ORC1-5 orthologs. Tree construction is based on established phylogeny (Misof et al, 2014) and distances depicted are arbitrary.

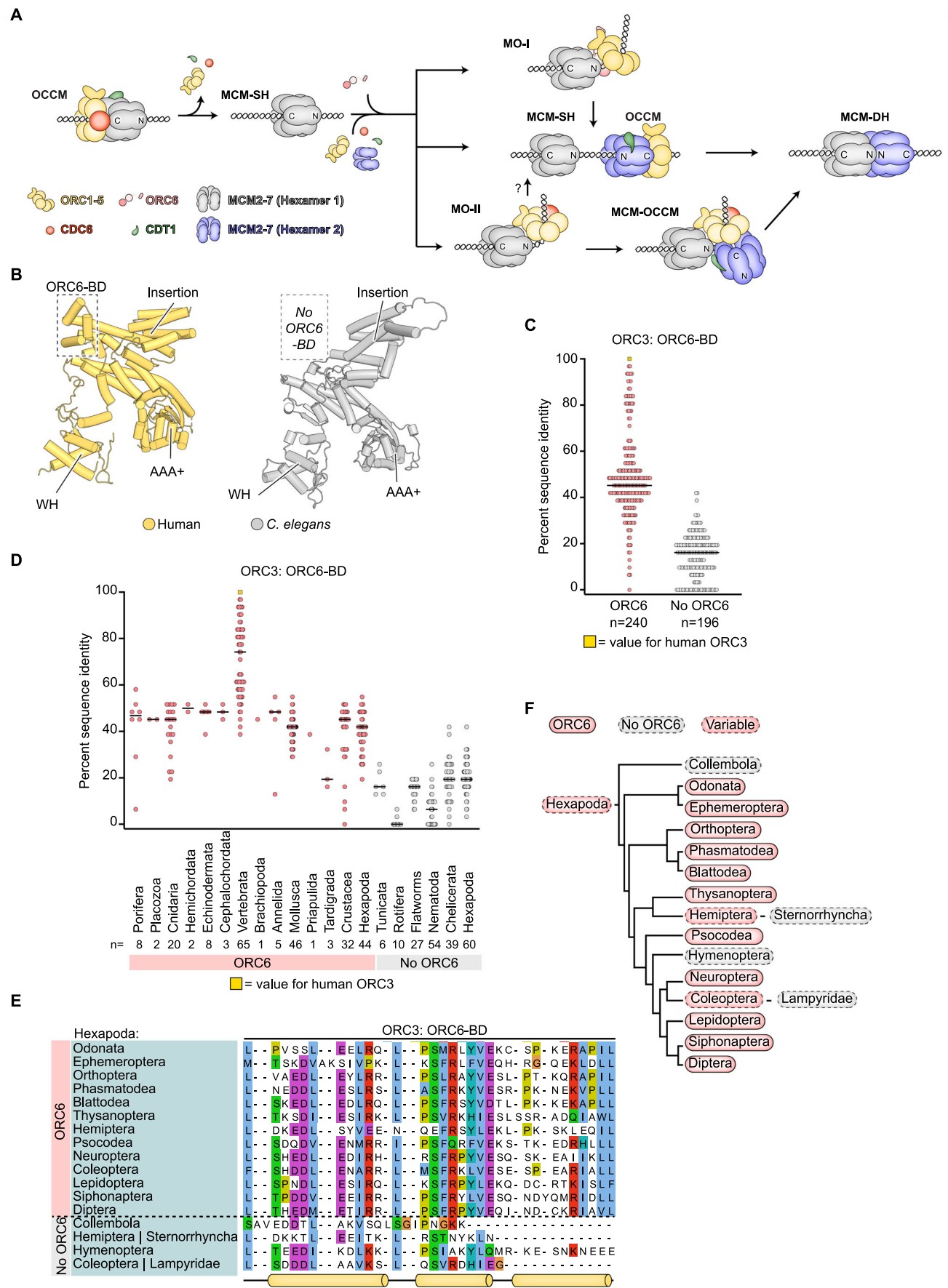

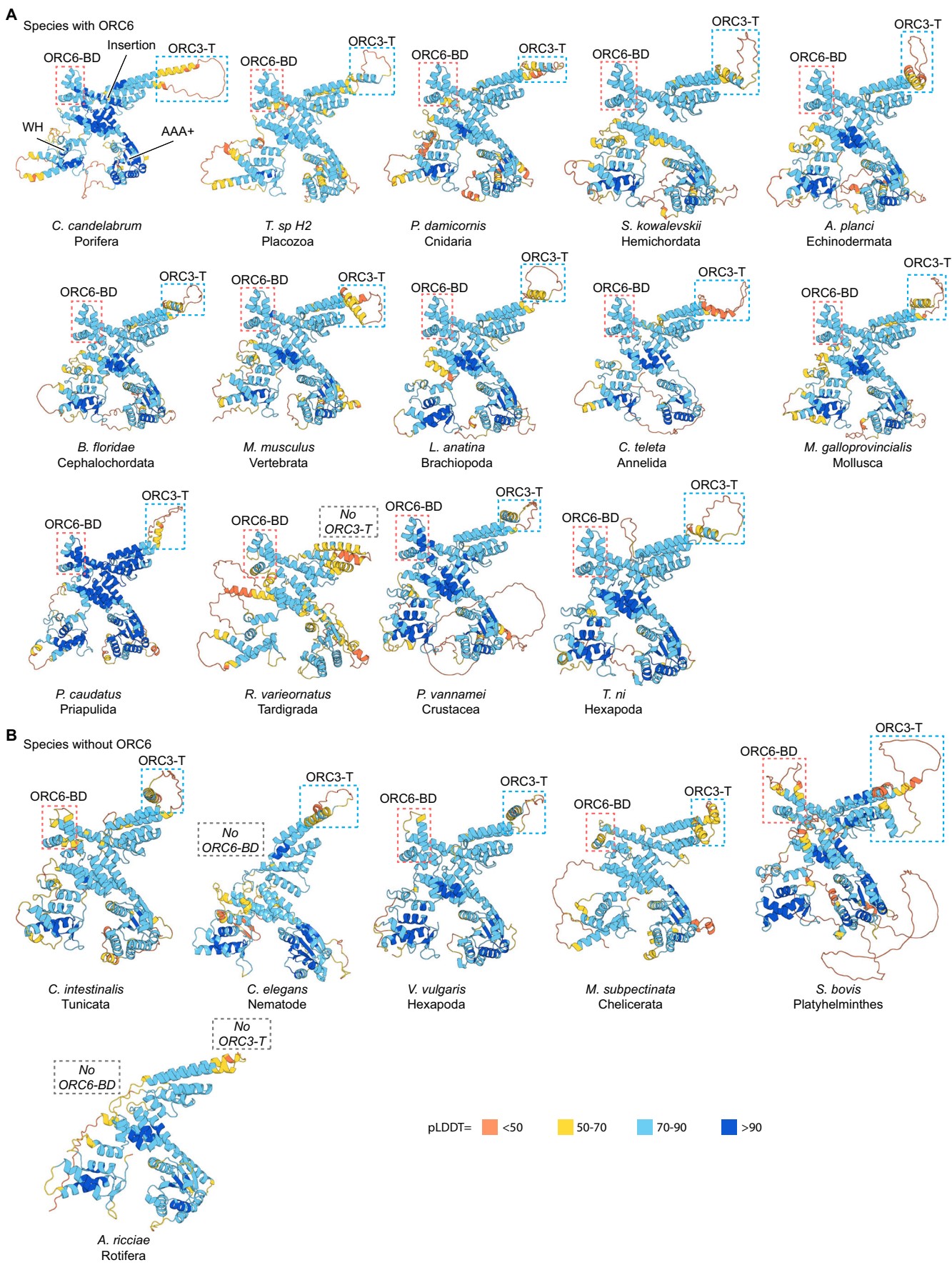

◀ **Figure EV2.  Example AlphaFold2 predictions of ORC3 orthologs.**

(**A**) ORC3 predictions for species with ORC6 and (**B**) without ORC6 orthologs for each phylum/subphylum in our dataset. Models are colored according to pLDDT scores. The ORC6-BD and ORC3 tether regions, if present, are labeled.

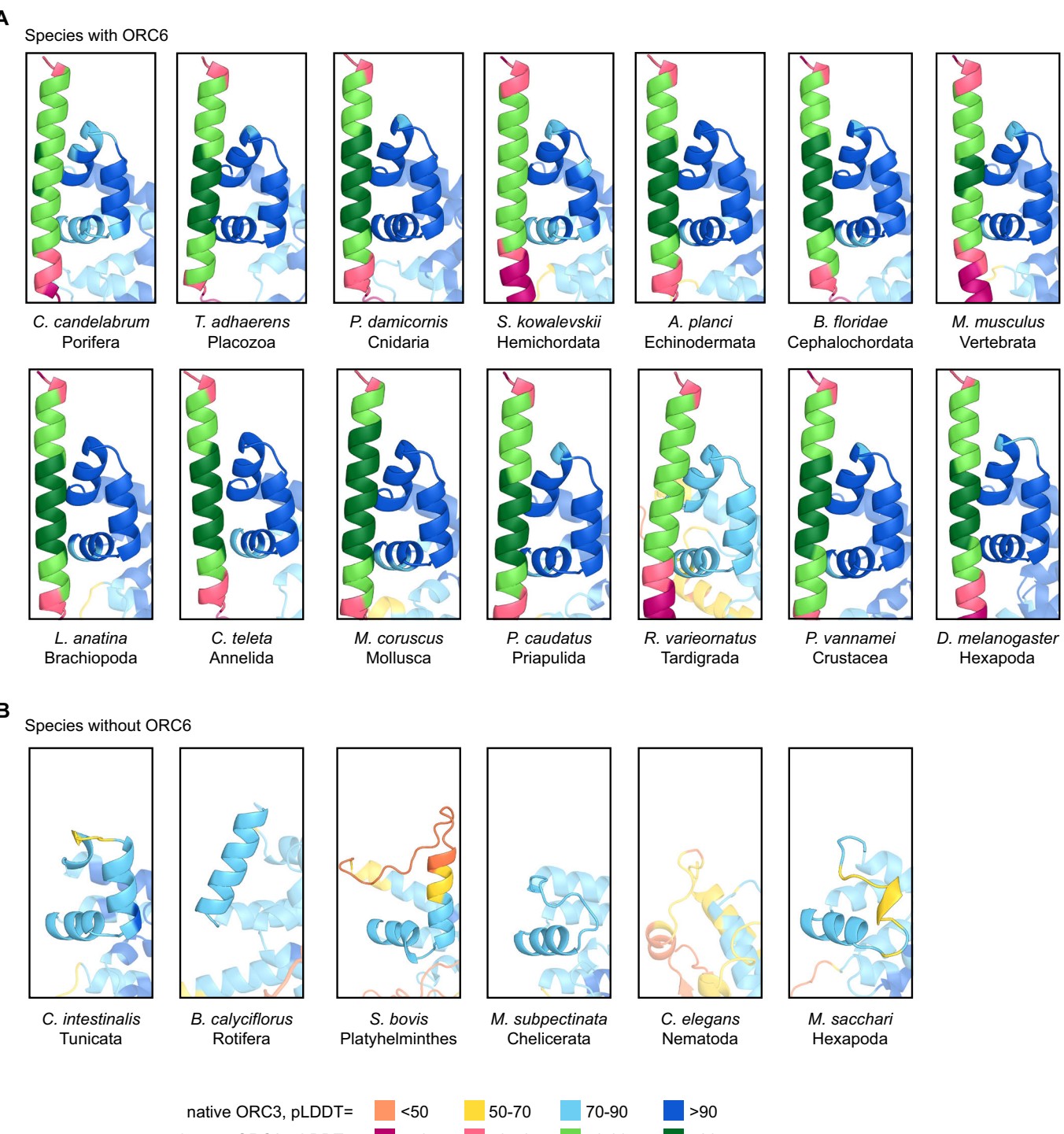

**A** Species with ORC6

*C. candelabrum*
Porifera

*T. adhaerens*
Placozoa

*P. damicornis*
Cnidaria

*S. kowalevskii*
Hemichordata

*A. planci*
Echinodermata

*B. floridae*
Cephalochordata

*M. musculus*
Vertebrata

*L. anatina*
Brachiopoda

*C. teleta*
Annelida

*M. coruscus*
Mollusca

*P. caudatus*
Priapulida

*R. varieornatus*
Tardigrada

*P. vannamei*
Crustacea

*D. melanogaster*
Hexapoda

**B** Species without ORC6

*C. intestinalis*
Tunicata

*B. calyciflorus*
Rotifera

*S. bovis*
Platyhelminthes

*M. subpectinata*
Chelicerata

*C. elegans*
Nematoda

*M. sacchari*
Hexapoda

native ORC3, pLDDT=  <50  50-70  70-90  >90
human ORC6, pLDDT=  <50  50-70  70-90  >90

**Figure EV3. Example AlphaFold2 Multimer-predicted interfaces between human ORC6 and ORC3 from various metazoan species.**

Example AlphaFold2 Multimer predictions between human ORC6 and ORC3 from species (**A**) with ORC6 and (**B**) without ORC6 orthologs for each phylum/subphylum in our dataset. Although all predictions were done with human ORC6 as input sequence, no interactions with ORC3 are predicted in species that have lost ORC6. Views are zoomed on the canonical ORC3-ORC6 interaction interface. Models are colored according to pLDDT scores.

**A**

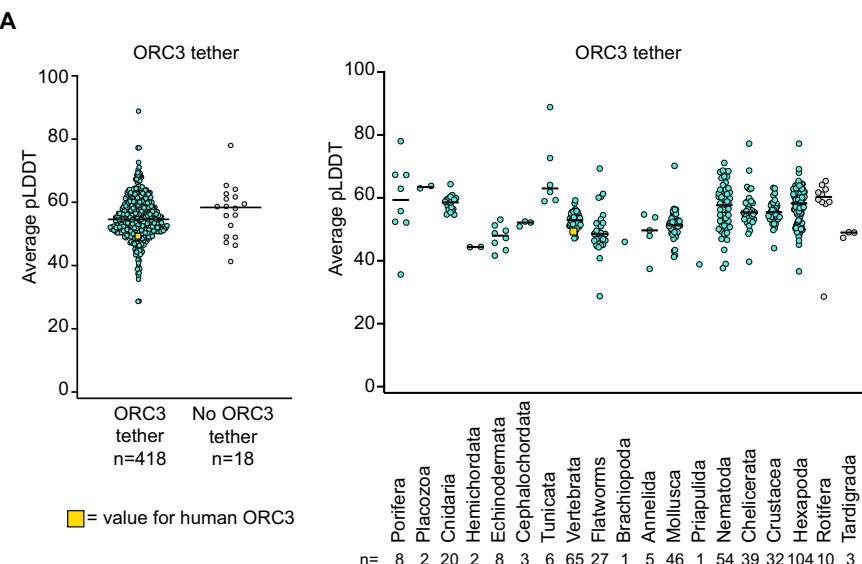

**B**

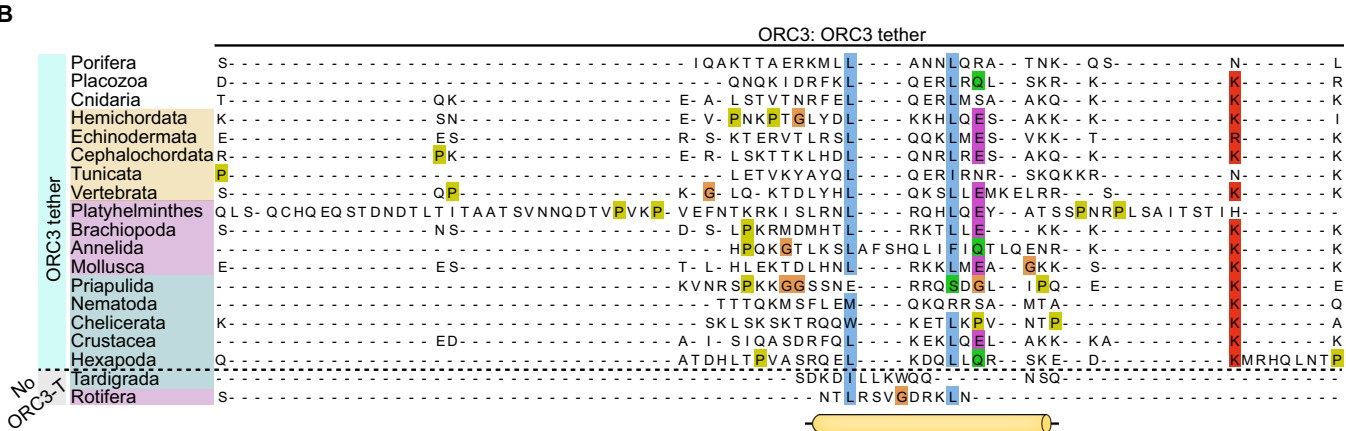

**Figure EV4. The ORC3 tether displays moderate structural and sequence conservation.**

(A) Average pLDDT scores for the ORC3 tether region in AlphaFold2 monomer predictions of ORC3 orthologs across Metazoa. Scores for ORC3 orthologs with and without the ORC3 tether are plotted on the left, and those for individual phyla/subphyla are plotted on the right. Solid black lines represent medians. (B) Multiple protein sequence alignment of ORC3 showing the ORC3 tether region. The cylinder below the alignment denotes a predicted α-helix in the tether.

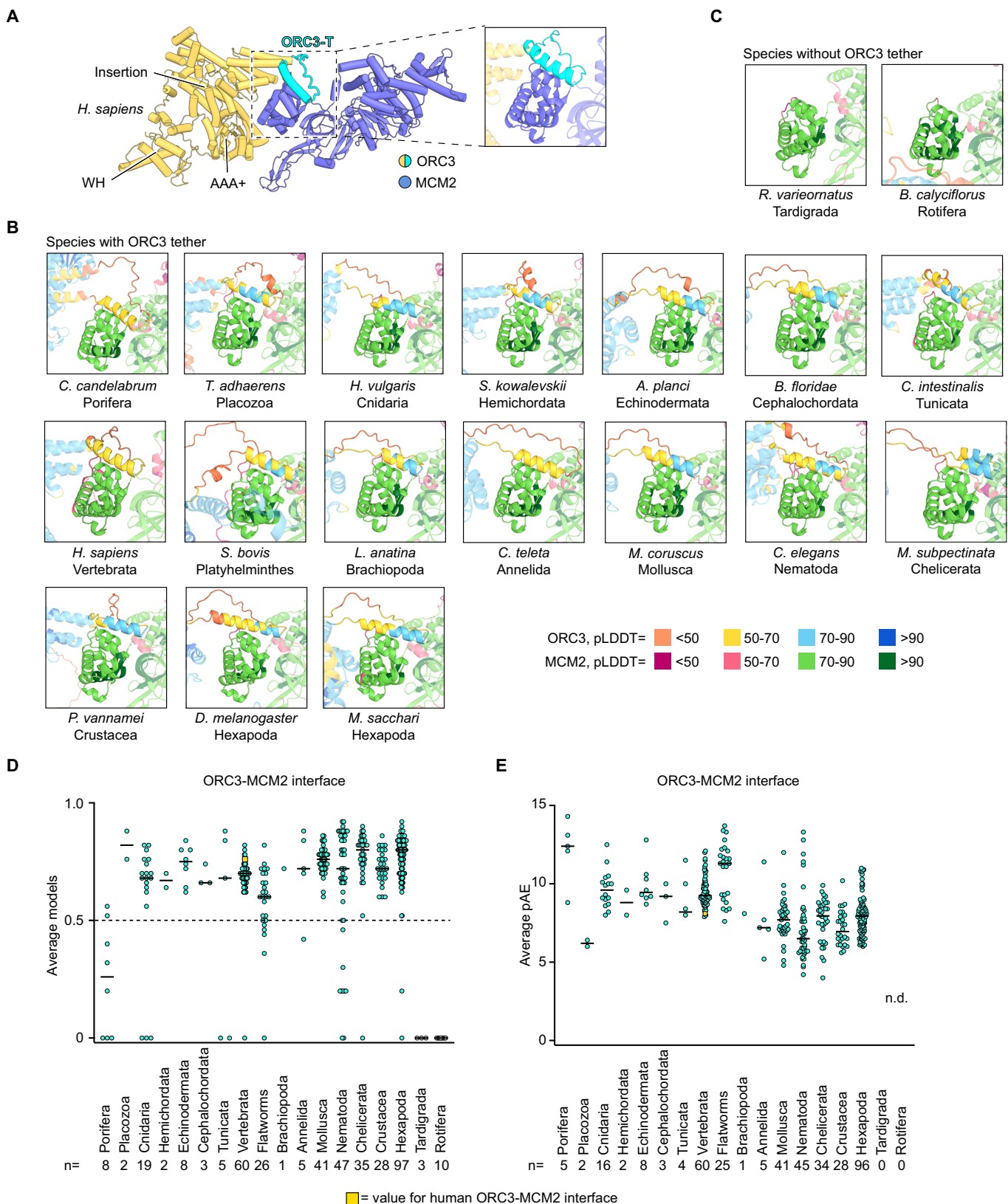

◀ **Figure EV5. Example AlphaFold2 multimer-predicted interfaces between metazoan ORC3 and MCM2 and associated confidence statistics.**

(A) AlphaFold2 multimer-predicted interactions between human ORC3 and MCM2. Zoomed view shows interaction between the MCM2 N-terminal domain and the ORC3 tether. (B, C) Example AlphaFold2 multimer-predicted ORC3-MCM2 interfaces for species with an ORC3 tether (in B) and without an ORC3 tether (in C). Models are colored according to pLDDT scores. In (C), no interaction between ORC3 and MCM2 is predicted by AlphaFold despite providing both sequences in the input. (D) *Average models* scores for AlphaFold2 multimer ORC3-MCM2 predictions by taxonomic group. Dotted black line at *average models* = 0.5 marks the confidence cut-off. (E) Average interface pAE scores of ORC3-MCM2 interactions by taxonomic group. If no interface was formed, pAE could not be calculated and no data is shown for these datapoints. Solid black lines in (D, E) are medians. Values of 0 in (D) indicate that no canonical ORC3-MCM2 interface was predicted.

