## [Peer Review File · The EMBO Journal]

AlphaFold-guided phylogenetic analyses suggest surprising heterogeneity in metazoan replication origin licensing mechanisms

Olivia Hunker and Franziska Bleichert

Corresponding author(s): Franziska Bleichert (franziska.bleichert@yale.edu)

Review Timeline:

Submission Date:	30th May 25
Editorial Decision:	21st Jul 25
Revision Received:	17th Sep 25
Editorial Decision:	15th Oct 25
Revision Received:	21st Oct 25
Accepted:	23rd Oct 25

Editor: Hartmut Vodermaier

Transaction Report:

Dr. Franziska Bleichert
Yale University
Molecular Biophysics and Biochemistry
260 Whitney Avenue
YSB 345
New Haven, CT 06520

21st Jul 2025

Re: EMBOJ-2025-121507-T
AlphaFold-guided phylogenetic analyses suggest surprising heterogeneity in metazoan origin licensing

Dear Dr. Bleichert,

Thank you again for submitting your phylogenetic analysis of metazoan ORC proteins and replication origins licensing for our consideration. Three expert referees have now looked at it, and provided the reports copied below. As you will see, all of them appreciate the interest of this work and its conclusions. Pending adequate revision of a number of specific concerns, we would therefore be happy to consider this work further for EMBO Journal publication.

As you will see, the reviewers raise mainly presentational points and requests for deeper computational follow-up on certain aspects. While referee 2 would additionally like to see experimental validation of certain conclusions, I feel that this well-taken request appears to go somewhat beyond the scope of the present investigation, and would alternatively only ask you to clearly state in abstract and discussion that the presented approach provides concrete hypotheses for follow-up experimentation.

When revising the manuscript for The EMBO Journal, please carefully note our guidelines and formatting conventions explained below and in our guide to authors, including the citation and referencing format for primary papers and preprints, the provision of separate text and figure files, and the various possibilities for including Expanded View Figures/Tables/Datasets. Adhering to these guidelines as closely as possible shall greatly facilitate the editorial process at the time of resubmission.

Thank you again for the opportunity to consider this work for The EMBO Journal, and I look forward to your revision in due time. Should you have any questions regarding your revision, please do not hesitate to contact me any time.

With kind regards,

Hartmut

9) To facilitate reproducibility and cross-laboratory adoption of methodologies, please structure the Materials & Methods section as outlined in our guide to authors, including a completed Reagents and Tools Table that can be downloaded from our author guidelines as well (<https://www.embopress.org/page/journal/14602075/authorguide#structuredmethods>).

10) Digital image enhancement is acceptable practice, as long as it accurately represents the original data and conforms to community standards. If a figure has been subjected to significant electronic manipulation, this must be clearly noted in the figure legend and/or the 'Materials and Methods' section. The editors reserve the right to request original versions of figures and the original images that were used to assemble the figure. Finally, we generally encourage uploading of numerical as well as gel/blot image source data; for details see: embopress.org/page/journal/14602075/authorguide#sourcedata

In the interest of ensuring the conceptual advance provided by the work, we recommend submitting a revision within 3 months (19th Oct 2025). Please discuss the revision progress ahead of this time with the editor if you require more time to complete the revisions. Use the link below to submit your revision:

Link Not Available

Referee #1:

The authors report an informative phylogenetic analysis powered by AlphaFold2 multimer. They show that several metazoan lineages have lost ORC6, a loading factor of the replicative helicase previously thought to be essential for origin licensing. Loss of ORC6 is coupled to the degeneration of the ORC6 binding domain of ORC3. In contrast, a recently identified ORC3 element, found to support helicase engagement during and DNA loading, is conserved across metazoans. Thus, this ORC6-independent pathway to helicase loading is likely used not only in humans but also in other metazoan species. Overall, the current study provides a useful perspective that aligns with a growing body of evidence indicating that multiple pathways coexist in cells to load the replicative helicase, in preparation to initiation of DNA replication.

Bigger issues

1. The authors searched the NCBI databases to find that ORC6 is absent in a subphylum of Chordata, Rotifera, flatworms, Nematoda, and a subphylum of Arthropoda. Have the authors considered that ORC6 might still exist in these organisms but be highly divergent (with the ORC6 DB of ORC3 evolving to follow this divergence)? It would be useful to address this issue in some form. If the authors can rule out this possibility, they should explain why.
2. The authors used AlphaFold2 predictions to make the point that a degenerate ORC6 DB in ORC3 would prevent binding of

ORC6 from related metazoan species. Given that both ORC1-5 as well as ORC6 bind duplex DNA, it would be useful to address whether concomitant DNA binding could help orient ORC6 and ORC1-5 in a productive fashion, despite the degenerate ORC6 binding domain in ORC3. Such modelling could be performed using AlphaFold3.

3. The authors find that ORC6 negative species are enriched in parasites. Can the authors comment on whether ORC6 from the host might be used in these species? Is this a scenario that the authors investigated using AlphaFold2?

Small issues:

1. "We inferred the conservation of each ORC subunit based on the presence and absence of orthologs across taxonomic groups."

Should be presence or absence instead?

2. The authors use phyla as a singular noun. However, phylum is singular and phyla is plural, if my memory serves me well.

Referee #2:

Hunker and Bleichert present timely new insight into the complexities of metazoan replication origin licensing. Recently, biochemical reconstitution of metazoan licensing has revealed that several molecular pathways can lead to loaded MCM double hexamers, the precursors of replication fork generation by origin firing. This was surprising because the better investigated yeast licensing system had not revealed such diverse separate pathways in this organism. The biological relevance of these different metazoan licensing pathways remained unaddressed. Hunker and Bleichert here connect to this remaining question. They seek a comprehensive picture from comparing licensing pathway-specific structural protein features in a wide range of metazoan phyla. A main method used is AlphaFold multimer-based complex formation prediction. They were able to reveal the differential presence of the two MO-dependent licensing pathways in metazoa, and propose a model in which these pathways act redundantly, allowing loss of one or the other in individual species. This develops a comprehensive picture of the two licensing pathways in the metazoan kingdom that would not be possible using conventional biochemical and structural approaches that inevitably focus on few individual species. Thus, the work presents a clever new way of using AlphaFold prediction with clear added value.

The computational analysis shown is thorough and overall convincing. The main limitation is that the key message of the study, that two pathways exist in metazoan cells, which act redundantly (half the discussion is about this point), remains biologically untested. I believe that addressing this issue is experimentally possible using current genetic and RNAi approaches in human tissue culture or in other genetic model organisms. Also biochemical binding studies could help.

Specific points to address:

1) Fig 2B/S1B:

- The labelling can be read as if the ORC6-DB is only the central helix, which can lead to a wrong impression about conservation. Please clarify the labelling to show that the whole region shown is the ORC6-DB. Also showing amino acids that form important interactions with ORC6 could be indicated for additional orientation.

- Nevertheless, some ORC6-BDs of ORC3 in organisms without ORC6 seem to match the human ORC6-BD quite well. Can statistical information regarding the degree of conservation in organisms with and without ORC6 be provided?

2) Fig 2e/3b/4b:

- A clear definition of the categories 'conserved' and 'non-conserved' ORC6-BD (2e) is missing.

- Because I expect that the categories of different metazoan phyla have different numbers of species genomes I think showing 'number of species' as a basis for the heat map is strange. A relative measure like % of species seems more appropriate.

3) Fig 3d:

The figure shows that the ORC6-ORC3 interfaces were predicted with a reasonably low probability of alignment error that is similar to the human ORC6-ORC3 prediction. However, the authors should take out the pAE values for species without ORC6, because the absence of an interface in most predictions with human ORC6 does not allow calculation of an pAE. Moreover, a pAE of nearly 0 that results from having no interface suggests a very good alignment (low pAE), which makes no sense if there is no interface to align. Consequently, no conclusion can be drawn from comparing the values for species with and without ORC6. The same argument applies for all similar pAE-involving results shown in other figures.

4) ORC6-ORC3 binding predictions:

For the targeted analysis of ORC6 binding to degenerate ORC6-BDs with low pLDDT, the authors should remove the pLDDT \geq 50 threshold from the algorithm to avoid a bias due to exclusion of residues with low pLDDT values in the ORC3-ORC6 interface. This would allow a better assessment whether or not degenerate ORC6-BDs can still bind ORC6.

5) line 296:

219 out of 240 ORC6-positive species show an interaction of ORC3 with human ORC6. A complementary argument would be if the remaining 21 ORC3s show an interaction with the ORC6 from their own species.

6) line 410:

'...small size of the interface...'

Is this small interface of the ORC3 tether likely to provide a high affinity interaction with Mcm2? Or does the tether cooperate with other low-affinity site for a high affinity interaction?

The model of functionally redundant interfaces for licensing (ORC3-ORC6 and ORC3-Mcm2) might benefit from a comparison of the interaction interfaces in terms of affinities. Is high affinity interaction required?

7) Concept of redundant ORC3 interfaces for licensing:

Indeed loss of one or the other licensing pathway (ORC6 and Mcm2-dependent) in some species argues for functional redundancy. However, redundancy is seen in cellular processes that require extreme reliability, such as the protection from re-replication by redundant CDK inhibition of mechanisms of licensing. Such reliability does not seem to be required for the basic licensing process. Does this rather speak for functional diversity rather than functional redundancy of the licensing processes?

8) Paragraph from line 485:

What is the domain structure of ORC3 in organisms that have lost ORC6 and ORC3 tether? Does it have any features in addition that could replace ORC6 or Orc3 tether function?

Minor issues:

Intro: A model of the licensing pathways would help, similar to Fig 5, perhaps as a supplementary figure.

line 370: please change to '...metazoans including non-human species.'

Referee #3:

A key event in genome duplication in eukaryotic organisms is the assembly of a head-to-head double hexamer of MCM2-7 complexes at origins of replication. Structural and biochemical evidence using human proteins indicates that after the first MCM2-7 complex is deposited around DNA, recruitment of the second MCM2-7 for DH formation can proceed via one of three distinct pathways, two of which are thought to involve specific protein-protein interactions. In pathway I, a c-terminal alpha helix in ORC6 docks onto an ORC6-binding domain in ORC3, called ORC6-BD, in MCM3. This interaction is believed to dock ORC onto the first MCM2-7 hexamer in such an orientation so that the second hexamer is recruited in the required inverse orientation. In pathway II, the ORC3 subunit of ORC binds the MCM2 subunit of MCM2-7 in a way that is analogous to the binding of ORC to MCM-7 in yeast, thereby coupling recruitment of the first and second MCM2-7 complexes in inverted orientations. However, how these pathways are utilized in different metazoans is unknown.

To address this, the Bleichert laboratory leveraged a "structural phylogenetics" approach in which they use sequence analysis to examine the presence or absence of particular genes or motifs in hundreds of metazoan species, and then use structure prediction to ask whether specific structural motifs and protein-protein interactions are predicted with high confidence. They first showed that most metazoan lineages contain a clear ORC6 homolog, but in those lacking ORC6, the ORC6-BD in ORC3 is also no longer present, consistent with these organisms truly lacking ORC6 (as opposed to an inability to detect the homolog). Moreover, in most organisms containing ORC6, the PPI between ORC6 and ORC3 is predicted with high confidence. The authors conclude that the ORC6-dependent pathway I of pre-RC formation was lost independently from several lineages. On the other hand, virtually all metazoans exhibit the interaction of ORC3-MCM2 interaction (pathway II) that mechanistically resembles the yeast pathway. Thus, pathway II represents the more fundamentally conserved mechanism. Interestingly, only rare organisms lack both pathways.

This paper presents a powerful approach to survey the presence of structural motifs and PPIs across a vast evolutionary space, which has, in this case, led to a compelling assessment of the relative importance of different mechanisms in different organisms. The presentation is clear, the evaluation of structure predictions is sound, and the conclusions are appropriate. My only significant criticism is that the interaction between ORC and MCM2-7 that underlies pathway I should be subjected to the same structural-phylogenetic interaction as the other PPIs surveyed in this paper. This is important, because the implication of pathway I as it stands is rather indirect. If the interaction is thought to involve the composite ORC3-ORC6 interface, this is not a barrier, as three-way interactions can readily be modeled with AlphaFold.

Specific Point

1. The scale in Figure 3e must be wrong. The PAE for the low confidence predictions cannot be 0. Same for S5B (and maybe other figures).

Response to reviewers

We thank the reviewers for their positive assessment of our work and suggestions that have allowed us to improve our manuscript. In this revision, we have addressed all reviewer comments, which included the following analyses:

- We have performed additional AlphaFold modeling to address the potential roles of ORC6-DNA interactions and MCM-ORC6 interactions in formation of the MO-I intermediate that are included in this response in **Figures R1** and **R2**.
- We have repeated the analysis of the ORC3-ORC6 interfaces without exclusion of low pLDDT residues to demonstrate that these criteria did not bias our results. The results are included as **Figure R3** in this response.
- We have calculated percent sequence identities for the ORC6-BD in ORC3. This new analysis is included in **Figures EV1C-D**.

In addition, we have made minor updates to the main text and several figures, as suggested by multiple reviewer comments. Changes in the manuscript text are marked using the track changes functionality in the enclosed word document. Page and line numbers in this response refer to the merged pdf document. Our point-by-point responses to the reviewers' comments are included below.

Referee 1:

The authors report an informative phylogenetic analysis powered by AlphaFold2 multimer. They show that several metazoan lineages have lost ORC6, a loading factor of the replicative helicase previously thought to be essential for origin licensing. Loss of ORC6 is coupled to the degeneration of the ORC6 binding domain of ORC3. In contrast, a recently identified ORC3 element, found to support helicase engagement during and DNA loading, is conserved across metazoans. Thus, this ORC6-independent pathway to helicase loading is likely used not only in humans but also in other metazoan species. Overall, the current study provides a useful perspective that aligns with a growing body of evidence indicating that multiple pathways coexist in cells to load the replicative helicase, in preparation to initiation of DNA replication.

We thank the reviewer for favorable assessment of our study.

Bigger issues

1. The authors searched the NCBI databases to find that ORC6 is absent in a subphylum of Chordata, Rotifera, flatworms, Nematoda, and a subphylum of Arthropoda. Have the authors considered that ORC6 might still exist in these organisms but be highly divergent (with the ORC6 DB of ORC3 evolving to follow this divergence)? It would be useful to address this issue in some form. If the authors can rule out this possibility, they should explain why.

We thank the reviewer for posing this question and have indeed considered this possibility. While we cannot definitively rule out this possibility, we think it is unlikely for several reasons: First, the CTD helix in ORC6 that binds to the ORC6-BD is a relatively small region of the ORC6 protein (~21 residues in human ORC6); therefore, even if the CTD of ORC6 and the ORC6-BD of ORC3 co-evolved, the cyclin box domains of ORC6 would retain significant homology to other ORC6

proteins and would likely remain detectable by sensitive homology-based search methods. Second, the ORC6-BD and surrounding region of the ORC3 insertion domain is not simply divergent but is completely absent in some nematode and rotifer ORC3 orthologs, therefore likely precluding the possibility of a co-evolved divergent ORC6-BD in ORC3 and ORC6-CTD in these species. Third, we are able to detect ORC6 in some groups within the hexapoda subphylum but not others (which is accompanied by a non-conserved ORC6-BD in ORC3). These ORC6-positive and ORC6-negative groups are more closely related to each other than other ORC6-positive phyla are to each other, so we would not expect that there has been sufficient time for the ORC6 and the ORC6-BD to co-evolve beyond recognition. Nonetheless, we have included a sentence in the manuscript (page 20, lines 439-441) to acknowledge this possibility.

2. The authors used AlphaFold2 predictions to make the point that a degenerate ORC6 DB in ORC3 would prevent binding of ORC6 from related metazoan species. Given that both ORC1-5 as well as ORC6 bind duplex DNA, it would be useful to address whether concomitant DNA binding could help orient ORC6 and ORC1-5 in a productive fashion, despite the degenerate ORC6 binding domain in ORC3. Such modelling could be performed using AlphaFold3.

We thank the reviewer for raising this issue, but we think such a scenario is unlikely.

First, experiments done both in our reconstituted human licensing system and in flies show that the cyclin box folds in ORC6 are not sufficient to support MCM loading or suffice to form the MO-I intermediate that is stabilized by ORC6. Even a single point mutation linked to Meier-Gorlin syndrome can interfere with ORC6/ORC6-BD interactions and impede MCM loading in both flies and humans¹⁻³. Thus, all three ORC6 domains must be present in *cis* to support ORC6's function in MCM loading.

Second, although ORC6 and its cyclin box folds in humans and flies have been reported to bind DNA, the affinity is extremely weak ($K_d > 10 \mu\text{M}^4$), ~3 orders of magnitude lower than for the core ORC1-5 complex ($K_d \sim 10\text{-}50 \text{ nM}^5$), suggesting ORC6 is unlikely to contribute to orienting ORC on DNA.

Third, *Drosophila* ORC6 cyclin box folds do not contact DNA or any other ORC subunits in several *Drosophila* ORC-DNA or ORC-DNA-CDC6 structures⁶.

Fourth, we have attempted to model this scenario using AlphaFold3 as suggested by the reviewer. We generated predictions of both human ORC6 in complex with DNA and the full ORC1-6 complex with DNA to determine if AlphaFold3 predicts interactions between these components that are consistent with the published MO structure⁷ to orient ORC on DNA. If ORC6-DNA interactions are able to facilitate MO formation in the absence of a canonical ORC3-ORC6 interaction, one would expect that AlphaFold3 predicted ORC6-DNA and ORC1-6-DNA complexes with interfaces that are consistent with the human MO structure. Comparison of all five AlphaFold3 models for the ORC6-DNA complex shows five different DNA binding modes, indicating low-confidence interfaces (**Figure R1A**). Moreover, AlphaFold3 models of ORC1-6 in complex with DNA show the ORC6 cyclin box domains of ORC6 packing against ORC1 and ORC4 and not against ORC4 and ORC5 as in the published MO structure⁷; as a result, ORC6 ends up on the opposite side of the DNA duplex (**Figure R1B**). Taken together, these AlphaFold3 predictions do not support a model where DNA binding of the cyclin box folds and ORC1-5 cooperate to orient ORC on DNA and are inconsistent with experimentally determined interfaces.

Finally, we would like to point out that the ORC6 cyclin box folds interact with highly conserved surfaces of the N-terminal domains of the MCM hexamer that are involved in MCM dimerization into the double hexamer. We anticipate that the evolutionary pressure to maintain MCM

dimerization would also exert evolutionary pressure on ORC6 to preserve interactions between the cyclin box folds and the MCM N-terminal domains for ORC6-dependent MCM loading. To test this concept, we used AlphaFold to predict interactions between human ORC6 and MCM2/MCM6 in ORC6-negative species; indeed, AlphaFold models maintained interactions in these cross-species predictions, consistent with ORC6's docking site for the cyclin box folds being preserved due to constraints on the MCM-NTDs (**Figure R2**). Thus, inability to identify ORC6's cyclin box folds in blast searches supports the absence of an ORC6 ortholog in these species.

Figure R1. AlphaFold3 modeling does not consistently predict interactions observed in the human MO structure. A) Comparison of five ORC6-DNA models generated by AlphaFold3 and the ORC6-DNA interaction observed in the human MO structure⁷. All models shown are aligned on the C-terminal cyclin-box fold of ORC6 (in salmon). **B)** Comparison of an AlphaFold3 model of human ORC1-6 in complex with DNA and the human MO structure. Structural models are aligned on the ORC1-5 complex. The BAH domain of ORC1 has been omitted in the AlphaFold3 model of ORC1-6 for visual clarity.

Figure R2. AlphaFold2 modeling between metazoan MCM2/MCM6 and human ORC6. A) Interaction between human ORC6 and human MCM2 and MCM6 in the human MO structure⁷. **B)** AlphaFold2 models of MCM2 and MCM6 of the indicated metazoan species and human ORC6. Although all metazoan species shown are missing an ORC6 ortholog, the ORC6 docking site on MCM is preserved as indicated by the predicted interaction between human ORC6 and these MCM2/MCM6. The disordered N-terminus of MCM2 has been omitted for visual clarity.

3. The authors find that ORC6 negative species are enriched in parasites. Can the authors comment on whether ORC6 from the host might be used in these species? Is this a scenario that the authors investigated using AlphaFold2?

We thank the reviewer for this suggestion. We have already investigated this possibility in the manuscript in AlphaFold Multimer predictions between human ORC6 and various metazoan species. Forty six of the ORC6-negative parasitic species included in our dataset are parasites that infect humans. The ORC3 homolog from all these species failed to form consistent interactions with human ORC6 in AlphaFold predictions, as shown in **Figures 3B-E** and **Appendix Figure S1**, indicating that host ORC6 is unlikely to be used as a source for ORC6 in these species. We would also like to note that not all parasites in the dataset were lacking ORC6. In particular, the ectoparasites *Pediculus humanus corporis* (human head louse) and *Ctenocephalides felis* (the cat flea), both have an ORC6 ortholog.

Small issues:

1. "We inferred the conservation of each ORC subunit based on the presence and absence of orthologs across taxonomic groups." Should be presence or absence instead?

We thank the reviewer for pointing out this error and have fixed it in the text.

2. The authors use phyla as a singular noun. However, phylum is singular and phyla is plural, if my memory serves me well.

The reviewer is correct, and we thank them for catching this grammatical error. We have fixed these instances in the text.

Referee 2:

Hunker and Bleichert present timely new insight into the complexities of metazoan replication origin licensing. Recently, biochemical reconstitution of metazoan licensing has revealed that several molecular pathways can lead to loaded MCM double hexamers, the precursors of replication fork generation by origin firing. This was surprising because the better investigated yeast licensing system had not revealed such diverse separate pathways in this organism. The biological relevance of these different metazoan licensing pathways remained unaddressed. Hunker and Bleichert here connect to this remaining question. They seek a comprehensive picture from comparing licensing pathway-specific structural protein features in a wide range of metazoan phyla. A main method used is AlphaFold multimer-based complex formation prediction. They were able to reveal the differential presence of the two MO-dependent licensing pathways in metazoa, and propose a model in which these pathways act redundantly, allowing loss of one or the other in individual species. This develops a comprehensive picture of the two licensing pathways in the metazoan kingdom that would not be possible using conventional biochemical and structural approaches that inevitably focus on few individual species. Thus, the work presents a clever new way of using AlphaFold prediction with clear added value.

We thank the reviewer for their positive assessment of our work.

The computational analysis shown is thorough and overall convincing. The main limitation is that the key message of the study, that two pathways exist in metazoan cells, which act redundantly

(half the discussion is about this point), remains biologically untested. I believe that addressing this issue is experimentally possible using current genetic and RNAi approaches in human tissue culture or in other genetic model organisms. Also biochemical binding studies could help.

We appreciate the reviewer's suggestion to test these hypotheses in a biological and biochemical context. While we enthusiastically agree that experimentally testing this hypothesis would be of great value, we feel such experimental work is beyond the scope of the current study, which focuses primarily on using computational methods to generate probable hypotheses and models for origin licensing across the metazoan kingdom.

We note that other studies have tested the role of ORC6 in origin licensing by knocking down ORC6 in human cell culture and failed to observe a significant reduction in chromatin-bound MCM, supporting the idea that ORC6 is not essential for origin licensing in humans^{8,9}. We would also like to note that studying the role of ORC6 *in vivo* via knockdown or knockout methods is non-trivial, as ORC6 has several well-established and essential non-licensing roles in humans and other metazoans^{8,10-12}. As such, ORC6 cannot be completely knocked out without severe consequences to cell viability, and any effects observed from an ORC6 knockdown must be carefully interpreted in the context of both licensing and non-licensing functions.

We are excited about the prospect of testing these hypotheses biochemically through *in vitro* reconstitution of divergent metazoan origin licensing pathways, as we have stated in our discussion. However, such an experiment would require cloning and purification of at least 13-14 individual polypeptides per species (depending on absence or presence of ORC6) and reconstitution of these proteins into a fully working origin licensing system in several metazoan systems. We believe this undertaking lies outside the scope of this project and would merit its own consideration. We have revised the abstract to acknowledge the limitations of our computational work.

Specific points to address:

1) Fig 2B/S1B:

- The labelling can be read as if the ORC6-DB is only the central helix, which can lead to a wrong impression about conservation. Please clarify the labelling to show that the whole region shown is the ORC6-DB. Also showing amino acids that form important interactions with ORC6 could be indicated for additional orientation.

We thank the reviewer for this suggestion and have updated the figure to improve the clarity of the labeling.

- Nevertheless, some ORC6-BDs of ORC3 in organisms without ORC6 seem to match the human ORC6-BD quite well. Can statistical information regarding the degree of conservation in organisms with and without ORC6 be provided?

We thank the reviewer for this suggestion and have added plots that show sequence-level percent identity conservation of the ORC6-BD of each species in the ORC3 dataset as compared to the human ORC6-BD in **Figures EV1C-D**. This additional analysis demonstrates that ORC6-BDs from organisms with ORC6 have a higher percent identity to the human ORC6-BD than ORC6-BDs from organisms without ORC6 and mirrors the structural conservation of the ORC6-BD.

2) Fig 2e/3b/4b:

- A clear definition of the categories 'conserved' and 'non-conserved' ORC6-BD (2e) is missing.

A definition of each of these categories was included in the methods section, in the subsection titled "Identification of conserved or non-conserved ORC6 binding domains and ORC3 tethers in ORC3 AlphaFold models", but we realize that this definition may not be easy to find. Therefore, we have added a note to the figure legends to point readers to this definition in the methods section to improve clarity.

- Because I expect that the categories of different metazoan phyla have different numbers of specie genomes I think showing 'number of species' as a basis for the heat map is strange. A relative measure like % of species seems more appropriate.

We thank the reviewer for this excellent suggestion and have revised the heat maps accordingly to show percent of species rather than the absolute number.

3) Fig 3d:

The figure shows that the ORC6-ORC3 interfaces were predicted with a reasonably low probability of alignment error that is similar to the human ORC6-ORC3 prediction. However, the authors should take out the pAE values for species without ORC6, because the absence of an interface in most predictions with human ORC6 does not allow calculation of an pAE. Moreover, a pAE of nearly 0 that results from having no interface suggests a very good alignment (low pAE), which makes no sense if there is no interface to align. Consequently, no conclusion can be drawn from comparing the values for species with and without ORC6. The same argument applies for all similar pAE-involving results shown in other figures.

We thank the reviewer for raising this issue and allowing us to clarify. As noted in the figure legend, these datapoints were originally meant to represent species for which no interface was formed, for which a pAE value could not be calculated. We understand how this representation may be confusing, especially as low pAE values typically indicate a confident interface prediction. For clarity, we have removed these datapoints from our average pAE plots in the revised manuscript.

4) ORC6-ORC3 binding predictions:

For the targeted analysis of ORC6 binding to degenerate ORC6-BDs with low pLDDT, the authors should remove the $pLDDT \geq 50$ threshold from the algorithm to avoid a bias due to exclusion of residues with low pLDDT values in the ORC3-ORC6 interface. This would allow a better assessment whether or not degenerate ORC6-BDs can still bind ORC6.

We thank the reviewer for raising this point. Residues with low pLDDT values were excluded from the analysis to avoid overinterpreting the AlphaFold2 models, as residues with low pLDDT values may be unreliable. To assure that this method has not introduced significant bias into the analysis, we have rerun the analysis without the pLDDT threshold. After removing this threshold, the average models scores remained largely unchanged (for species with ORC6, the median becomes 0.92 as compared to 0.9; for species without ORC6 the median remains 0), suggesting that very few interface residues were excluded due to the pLDDT threshold (**Figure R3**). Only one ORC6-negative species, a flatworm, had an average models score increase above 0.5, and this interface was likely spurious as it consisted of only one contact between the flatworm ORC3 and human ORC6. A comparison of the average models scores with and without the pLDDT threshold is included below in **Figure R3**.

Figure R3. Including or excluding low pLDDT threshold residues when calculating average models score of the ORC3-ORC6 interface does not significantly change results. Left: Average models scores for ORC3-ORC6 AlphaFold2 Multimer predictions between metazoan ORC3 orthologs and human ORC6, for species with and without ORC6, with residues with $pLDDT \leq 50$ excluded from analysis, as shown in main text figure panel 3C. Right: Average models scores for ORC3-ORC6 AlphaFold2 Multimer predictions between metazoan ORC3 orthologs and human ORC6, for species with and without ORC6, with residues with $pLDDT \leq 1$ excluded from analysis (note that pLDDT cannot be set to 0).

5) line 296:

219 out of 240 ORC6-positive species show an interaction of ORC3 with human ORC6. A complementary argument would be if the remaining 21 ORC3s show an interaction with the ORC6 from their own species.

The reviewer raises an excellent point. This comment prompted us to re-examine this datapoint, which revealed a typo in this line; it was actually 220 out of 240 ORC6-positive species that showed an interaction with human ORC6, and 20 species that did not. This has now been corrected in the manuscript.

Native ORC6 protein sequences are not available for all species in our ORC6-positive dataset and therefore precludes running this analysis. To clarify why these outliers have been categorized as an ORC6-positive species, we began this study with the assumption that all animals possess a canonical ORC6 ortholog, as ORC6 was previously thought to be an essential origin licensing protein, and placed the burden of proof on proving that ORC6 is absent. These outliers did not meet that burden of proof, so we have conservatively categorized them with the ORC6-positive group. As noted in the manuscript, we did not infer the presence or absence of ORC6 at the species level because some genome assemblies are incomplete. This group of 20 species include two vertebrates, six crustaceans, four hexapods, two cnidarians, one echinoderm, three mollusks, one sponge, and one annelid, all of which are closely related to multiple species for which we did positively identify an ORC6 ortholog.

There are several possible explanations for these outliers. One possibility is that some of these species may in fact lack an ORC6 ortholog, and loss of ORC6 could be more widespread across the animal kingdom than reported here. Another possible explanation is some of these cases may represent contaminated genome sequencing samples, which may cause a sequence to be mislabeled as an incorrect species. As the goal of this experiment was to further interrogate whether taxonomic groups that are entirely lacking ORC6 orthologs are truly missing ORC6, we did not extensively investigate individual species within ORC6-positive taxonomic groups for which there was inconclusive evidence on whether they possess an ORC6 ortholog or not.

6) line 410:

'...small size of the interface...'

Is this small interface of the ORC3 tether likely to provide a high affinity interaction with Mcm2? Or does the tether cooperate with other low-affinity site for a high affinity interaction? The model of functionally redundant interfaces for licensing (ORC3-ORC6 and ORC3-Mcm2) might benefit from a comparison of the interaction interfaces in terms of affinities. Is high affinity interaction required?

This is a great question, and we thank the reviewer for prompting further discussion. Precise values for the affinities of both interfaces, as well as exact concentrations of ORC and MCM within the nucleus, which would be required to interpret the biological significance of these affinities, are unknown. We feel such experiments lie outside the scope of this study, but this would be a great question to pose as a follow-up.

We'd also like to point out that ORC3-ORC6 interface is rather small and of comparable size to the ORC3-MCM2 interface. Of note, ORC3-ORC6 affinities vary among metazoans. Human ORC6 does not stably associate with human ORC1-5 and does not co-purify with the ORC1-5 components^{2,13-15}. However, in *Drosophila melanogaster*, ORC6 binds ORC3 with low nanomolar affinity and does stably associate and co-purify with ORC1-5^{2,16}.

7) Concept of redundant ORC3 interfaces for licensing:

Indeed loss of one or the other licensing pathway (ORC6 and Mcm2-dependent) in some species argues for functional redundancy. However, redundancy is seen in cellular processes that require extreme reliability, such as the protection from re-replication by redundant CDK inhibition of mechanisms of licensing. Such reliability does not seem to be required for the basic licensing process. Does this rather speak for functional diversity rather than functional redundancy of the licensing processes?

The reviewer raises an interesting point and is correct that one could also characterize the described licensing pathways as an example of functional diversity in the case where some pathways are found only in some branches of the metazoan tree.

We respectfully argue that this is also an example of functional redundancy as we have discussed, and that the two concepts are not mutually exclusive. While there may be more margin for error in origin licensing compared to a process like preventing re-replication, we would argue that origin licensing as a whole does require a high degree of reliability in the sense that it is an essential cellular process, a critical threshold of origin licensing must be obtained in order for replication to proceed successfully, and a checkpoint is triggered if this is not achieved^{17,18}. While the origin

licensing system may not be regularly challenged during a typical cell cycle, it may be pushed to its limits during times of rapid cellular division, such as in development programs of complex multicellular organisms. In these scenarios, the necessity of redundant origin licensing pathways may become more applicable, and this hypothesis is supported by the disruption of development in Meier-Gorlin syndrome and its association with mutations in ORC6.

8) Paragraph from line 485:

What is the domain structure of ORC3 in organisms that have lost ORC6 and ORC3 tether? Does it have any features in addition that could replace ORC6 or Orc3 tether function?

The reviewer raises an excellent question. The overall domain architecture of ORC3 does not differ among ORC6-positive and ORC6-negative species (see **Figures EV1B and EV2** and the discussion on page 10, lines 213-216 in the manuscript). The only significant difference observed is that the ORC3 insertion domain is truncated near the ORC6-BD region in some nematode and rotifer species. While we cannot definitively rule out this possibility, we have not observed insertions in ORC3 in any ORC6-negative species that are predicted to form ordered domains.

Minor issues:

Intro: A model of the licensing pathways would help, similar to Fig 5, perhaps as a supplementary figure.

We thank the reviewer for this suggestion and have included a model of the licensing pathways as **Figure EV1A**.

line 370: please change to '...metazoans including non-human species.'

We have updated the text accordingly.

Referee 3:

A key event in genome duplication in eukaryotic organisms is the assembly of a head-to-head double hexamer of MCM2-7 complexes at origins of replication. Structural and biochemical evidence using human proteins indicates that after the first MCM2-7 complex is deposited around DNA, recruitment of the second MCM2-7 for DH formation can proceed via one of three distinct pathways, two of which are thought to involve specific protein-protein interactions. In pathway I, a c-terminal alpha helix in ORC6 docks onto an ORC6-binding domain in ORC3, called ORC6-BD, in MCM3. This interaction is believed to dock ORC onto the first MCM2-7 hexamer in such an orientation so that the second hexamer is recruited in the required inverse orientation. In pathway II, the ORC3 subunit of ORC binds the MCM2 subunit of MCM2-7 in a way that is analogous to the binding of ORC to MCM-7 in yeast, thereby coupling recruitment of the first and second MCM2-7 complexes in inverted orientations. However, how these pathways are utilized in different metazoans is unknown.

To address this, the Bleichert laboratory leveraged a "structural phylogenetics" approach in which they use sequence analysis to examine the presence or absence of particular genes or motifs in hundreds of metazoan species, and then use structure prediction to ask whether specific structural motifs and protein-protein interactions are predicted with high confidence. They first showed that most metazoan lineages contain a clear ORC6 homolog, but in those lacking ORC6,

the ORC6-BD in ORC3 is also no longer present, consistent with these organisms truly lacking ORC6 (as opposed to an inability to detect the homolog). Moreover, in most organisms containing ORC6, the PPI between ORC6 and ORC3 is predicted with high confidence. The authors conclude that the ORC6-dependent pathway I of pre-RC formation was lost independently from several lineages. On the other hand, virtually all metazoans exhibit the interaction of ORC3-MCM2 interaction (pathway II) that mechanistically resembles the yeast pathway. Thus, pathway II represents the more fundamentally conserved mechanism. Interestingly, only rare organisms lack both pathways.

This paper presents a powerful approach to survey the presence of structural motifs and PPIs across a vast evolutionary space, which has, in this case, led to a compelling assessment of the relative importance of different mechanisms in different organisms. The presentation is clear, the evaluation of structure predictions is sound, and the conclusions are appropriate. My only significant criticism is that the interaction between ORC and MCM2-7 that underlies pathway I should be subjected to the same structural-phylogenetic interaction as the other PPIs surveyed in this paper. This is important, because the implication of pathway I as it stands is rather indirect. If the interaction is thought to involve the composite ORC3-ORC6 interface, this is not a barrier, as three-way interactions can readily be modeled with AlphaFold.

We thank the reviewer for their positive feedback and appreciation of our work. The reviewer raises an important observation, that in the MO-I intermediate of this pathway, ORC6 makes contacts with MCM2 and MCM6 in addition to ORC3. We chose to focus our analysis on the ORC3-ORC6 interface and did not include an analysis of the ORC6-MCM interface for several reasons, and we thank the reviewer for the opportunity to explain our reasoning.

As we stated in our response to Reviewer 1, our previously published biochemical data suggests that the ORC3-ORC6 interaction is crucial for efficient formation of the MO-I intermediate. We observed a requirement for all ORC6 domains to be present *in cis* for MO-I formation, as well as a significant abrogation of MO formation upon a single point mutation to the ORC6 CTD which binds ORC3¹. These data suggest that the interaction between ORC6 and ORC3 is key for MO-I formation.

Furthermore, the ORC6-BD region of ORC3 was an ideal candidate for our phylogenetic analysis because its only known function is to bind to ORC6, predicting that ORC6 exerts sole influence on ORC6-BD conservation. In contrast, the N-termini of MCM2 and MCM6 that bind to ORC6 in the MO-I also play an essential role in forming the MCM double hexamer interface, which is expected to place strong evolutionary pressure on maintaining the MCM-NTDs to support MCM dimerization. Consequently, interactions of human ORC6 with these MCM-NTDs do not argue for presence of ORC6 in ORC6-negative species but likely reflect conservation of the MCM double hexamer structure. Consistent with this notion, AlphaFold predicts interactions between human ORC6 and native MCM2/6 in ORC6-negative species that we have tested (**Figure R2**, see response to point 2 of Reviewer 1). As conservation of the MCM2 and MCM6 N-termini would not provide evidence of its involvement in establishing the MO-I intermediate, nor would conservation of an MCM-ORC6 interface provide evidence for or against the conservation of ORC6 across the metazoan kingdom, we feel a structural and sequence level conservation analysis of the NTD regions of MCM2 and MCM6 would not be informative with regards to the MCM-ORC6 interaction.

Specific Point

1. The scale in Figure 3e must be wrong. The PAE for the low confidence predictions cannot be

0. Same for S5B (and maybe other figures).

We thank the reviewer for pointing this out. As explained in our response to point 3 of Reviewer 2, these datapoints were originally meant to represent species for which no interface was formed and for which a pAE could not be calculated, as was noted in the figure legends. We understand how this representation may be confusing, especially for readers that are familiar with the pAE scale. For clarity, we have removed these datapoints from our average pAE plots.

References

- 1 Yang, R., Hunker, O., Wise, M. & Bleichert, F. Multiple mechanisms for licensing human replication origins. *Nature* **636**, 488-498 (2024).
- 2 Bleichert, F. *et al.* A Meier-Gorlin syndrome mutation in a conserved C-terminal helix of Orc6 impedes origin recognition complex formation. *eLife* **2**, e00882 (2013).
- 3 Balasov, M., Akhmetova, K. & Chesnokov, I. Drosophila model of Meier-Gorlin syndrome based on the mutation in a conserved C-Terminal domain of Orc6. *Am J Med Genet A* **167A**, 2533-2540 (2015).
- 4 Xu, N. *et al.* Structural basis of DNA replication origin recognition by human Orc6 protein binding with DNA. *Nucleic Acids Res* **48**, 11146-11161 (2020).
- 5 Bleichert, F., Leitner, A., Aebersold, R., Botchan, M. R. & Berger, J. M. Conformational control and DNA-binding mechanism of the metazoan origin recognition complex. *Proc Natl Acad Sci U S A* **115**, E5906-E5915 (2018).
- 6 Schmidt, J. M. & Bleichert, F. Structural mechanism for replication origin binding and remodeling by a metazoan origin recognition complex and its co-loader Cdc6. *Nat Commun* **11**, 4263 (2020).
- 7 Weissmann, F. *et al.* MCM double hexamer loading visualized with human proteins. *Nature* **636**, 499-508 (2024).
- 8 Lin, Y. C. *et al.* Orc6 is a component of the replication fork and enables efficient mismatch repair. *Proc Natl Acad Sci U S A* **119**, e2121406119 (2022).
- 9 Hayashi-Takanaka, Y., Hiratani, I., Haraguchi, T. & Hiraoka, Y. Proteasome-dependent Orc6 removal from chromatin upon S-phase entry safeguards against minichromosome maintenance complex reloading and tetraploidy. *J Cell Sci* **138** (2025).
- 10 Prasanth, S. G., Prasanth, K. V. & Stillman, B. Orc6 involved in DNA replication, chromosome segregation, and cytokinesis. *Science* **297**, 1026-1031 (2002).
- 11 Chesnokov, I. N., Chesnokova, O. N. & Botchan, M. A cytokinetic function of Drosophila ORC6 protein resides in-a domain distinct from its replication activity. *Proceedings of the National Academy of Sciences of the United States of America* **100**, 9150-9155 (2003).
- 12 Bernal, J. A. & Venkitaraman, A. R. A vertebrate N-end rule degron reveals that Orc6 is required in mitosis for daughter cell abscission. *J Cell Biol* **192**, 969-978 (2011).
- 13 Dhar, S. K., Delmolino, L. & Dutta, A. Architecture of the human origin recognition complex. *J Biol Chem* **276**, 29067-29071 (2001).
- 14 Dhar, S. K. & Dutta, A. Identification and characterization of the human ORC6 homolog. *J Biol Chem* **275**, 34983-34988 (2000).

- 15 Vashee, S., Simancek, P., Challberg, M. D. & Kelly, T. J. Assembly of the human origin recognition complex. *J Biol Chem* **276**, 26666-26673 (2001).
- 16 Chesnokov, I., Gossen, M., Remus, D. & Botchan, M. Assembly of functionally active *Drosophila* origin recognition complex from recombinant proteins. *Genes Dev* **13**, 1289-1296 (1999).
- 17 Nevis, K. R., Cordeiro-Stone, M. & Cook, J. G. Origin licensing and p53 status regulate Cdk2 activity during G(1). *Cell Cycle* **8**, 1952-1963 (2009).
- 18 Shreeram, S., Sparks, A., Lane, D. P. & Blow, J. J. Cell type-specific responses of human cells to inhibition of replication licensing. *Oncogene* **21**, 6624-6632 (2002).

Dr. Franziska Bleichert
Yale University
Molecular Biophysics and Biochemistry
260 Whitney Avenue
YSB 345
New Haven, CT 06520

15th Oct 2025

Re: EMBOJ-2025-121507R

AlphaFold-guided phylogenetic analyses suggest surprising heterogeneity in metazoan replication origin licensing mechanisms

Dear Dr. Bleichert,

Thank you for submitting your revised manuscript to The EMBO Journal. It has now been re-reviewed by original referees 1 and 2, who were both satisfied with the revisions in response to their specific concerns. While referee 2 still would like to see experimental follow-up, we would -as stated in my original decision letter- we continue to consider this beyond to scope of the present computational approach for model generation, and shall therefore be happy to accept the study.

Prior to that, there are still some editorial issues that need to be addressed:

- Please adjust the order of the manuscript sections, and also make sure to use the correct section headers: Title page with complete author information, Abstract, Keywords, Introduction, Results, Discussion, Methods, Data Availability, Acknowledgements, Disclosure and Competing Interests Statement, References, Main Figure Legends, Tables, Expanded Figure Legends.
- On the abstract page of the manuscript, please include 4-5 general keyword terms to enhance searchability.
- Please carefully go through the reference list, which currently contains many wrongly formatted entries:
 - * some citations are incomplete, missing e.g. citation year, volume, and page/locator numbers
 - * Please adjust the format for citation of real preprints: The citation in the text should be: "(preprint: NAME1 et al, YEAR)"; and in the reference list: "NAME1, NAME2, ... (YEAR) article title. BIORXIV doi: XXX"
- Please include a Disclosure and competing interests statement (next to the Acknowledgment section) - for details, see <https://www.embopress.org/competing-interests>
- As we are switching from a free-text author contribution statement towards a more formal statement based on Contributor Role Taxonomy (CRediT) terms, please remove the present Author Contribution section and instead specify each author's contribution(s) directly in the Author Information page of our submission system during upload of the final manuscript. See <https://casrai.org/credit/> for more information.
- During routine pre-acceptance checks, our data editors noted that the exact p-values are not provided in the legends of figures 2F, 3C-E; 4D - please add them.
- Given that the current Expanded View Tables EV2-4 are extensive datasets, please rename them into "Dataset EV1-3", and update the in-text call-outs accordingly. Also, please remove the EV Table/Dataset legends from the main text, they should just be retained as separate tab on the respective XLSX spreadsheet, but again with nomenclature updated from Suppl. Table to "Dataset EV1/2/3".
- I would suggest to make the title somewhat more explicit and broadly appealing by including a reference to "replication", e.g. "AlphaFold-guided phylogenetic analyses suggest surprising heterogeneity in metazoan replication origin licensing mechanisms"
- Finally, please provide suggestions for a short 'blurb' text prefacing and summing up the conceptual aspect of the study in two sentences (max. 250 characters), followed by 3-5 one-sentence 'bullet points' with brief factual statements of key results of the paper; they will form the basis of an editor-written 'Synopsis' accompanying the online version of the article. Please also upload a synopsis image, which can be used as a "visual title" for the synopsis section of your paper. The image should be in PNG or JPG format, and please make sure that it remains in the modest dimensions of (exactly) 550 pixels wide and 300-600 pixels high. Figure 5 could serve as a basis, but it would have to be considerably simpler.

I am returning the manuscript to you for a final round of minor revision, solely to allow you to make these modifications and upload the revised files. Once we will have received them, we should be ready to swiftly proceed with formal acceptance and

production of the manuscript.

With kind regards,

Hartmut

- 1) Every manuscript requires a Data Availability section (even if only stating that no deposited datasets are included). Primary datasets or computer code produced in the current study have to be deposited in appropriate public repositories prior to resubmission, and reviewer access details provided in case that public access is not yet allowed. Further information: embopress.org/page/journal/14602075/authorguide#dataavailability
- 2) Each figure legend must specify
 - size of the scale bars that are mandatory for all micrograph panels
 - the statistical test used to generate error bars and P-values
 - the type error bars (e.g., S.E.M., S.D.)
 - the number (n) and nature (biological or technical replicate) of independent experiments underlying each data point
 - Figures may not include error bars for experiments with $n < 3$; scatter plots showing individual data points should be used instead.
- 3) Revised manuscript text (including main tables, and figure legends for main and EV figures) has to be submitted as editable text file (e.g., .docx format). We encourage highlighting of changes (e.g., via text color) for the referees' reference.
- 4) Each main and each Expanded View (EV) figure should be uploaded as individual production-quality files (preferably in .eps, .tif, .jpg formats). For suggestions on figure preparation/layout, please refer to our Figure Preparation Guidelines: <http://bit.ly/EMBOPressFigurePreparationGuideline>
- 5) Point-by-point response letters should include the original referee comments in full together with your detailed responses to them (and to specific editor requests if applicable), and also be uploaded as editable (e.g., .docx) text files.
- 6) Please complete our Author Checklist, and make sure that information entered into the checklist is also reflected in the manuscript; the checklist will be available to readers as part of the Review Process File. A download link is found at the top of our Guide to Authors: embopress.org/page/journal/14602075/authorguide
- 7) All authors listed as (co-)corresponding need to deposit, in their respective author profiles in our submission system, a unique ORCID identifier linked to their name. Please see our Guide to Authors for detailed instructions.
- 8) Please note that supplementary information at EMBO Press has been superseded by the 'Expanded View' for inclusion of additional figures, tables, movies or datasets; with up to five EV Figures being typeset and directly accessible in the HTML version of the article. For details and guidance, please refer to: embopress.org/page/journal/14602075/authorguide#expandedview
- 9) To facilitate reproducibility and cross-laboratory adoption of methodologies, please structure the Materials & Methods section as outlined in our guide to authors, including a completed Reagents and Tools Table that can be downloaded from our author guidelines as well (<https://www.embopress.org/page/journal/14602075/authorguide#structuredmethods>).
- 10) Digital image enhancement is acceptable practice, as long as it accurately represents the original data and conforms to community standards. If a figure has been subjected to significant electronic manipulation, this must be clearly noted in the figure legend and/or the 'Materials and Methods' section. The editors reserve the right to request original versions of figures and the original images that were used to assemble the figure. Finally, we generally encourage uploading of numerical as well as gel/blot image source data; for details see: embopress.org/page/journal/14602075/authorguide#sourcedata

In the interest of ensuring the conceptual advance provided by the work, we recommend submitting a revision within 3 months (13th Jan 2026). Please discuss the revision progress ahead of this time with the editor if you require more time to complete the revisions. Use the link below to submit your revision:

Link Not Available

Referee #1:

The authors have adequately addressed the issues I raised.

Referee #2:

The revisions presented have improved the manuscript's clarity and precision. It is now even clearer than it was before and provides an important new broad perspective about the licensing pathways that may be used in the metazoan kingdom. Unfortunately, my main conceptual concern regarding functional insight to back the computational predictions has not been addressed, which, I believe, leaves main conclusions in the somewhat speculative, albeit perhaps likely, realm. I think the manuscript in its current form is a better fit for a computational or structural journal.